

# Parameterization of Single Scattering Albedo (SSA) and Absorption Angstrom Exponent (AAE) with EC/OC for Aerosol Emissions from Biomass Burning

Rudra P. Pokhrel[1], Nick L. Wagner[2], Justin M. Langridge[3], Daniel A. Lack[4], Thilina Jayarathne[5],
Elizabeth A. Stone[5], Chelsea E. Stockwell[6], Robert J. Yokelson[6] and Shane M. Murphy[1]

[1]Department of Atmospheric Science, University of Wyoming, Laramie, Wyoming, USA
[2]NOAA Earth System Research Laboratory, Chemical Sciences Division, Boulder, Colorado, USA
[3]Observation Based Research, Met Office, Fitzroy Road, Exeter, EX1 3PB, UK
[4]Transport Emissions, Air Quality and Climate Consulting, Brisbane, Australia
[5]Department of Chemistry, University of Iowa, Iowa City, Iowa, USA
[6]Department of Chemistry, University of Montana, Missoula, Montana, USA

*Correspondence* to: S. M. Murphy (Shane.Murphy@uwyo.edu)

**Abstract.** Single scattering albedo (SSA) and absorption angstrom exponent (AAE) are two critical parameters in determining the impact of absorbing aerosol on the Earth's radiative balance. Aerosol emitted by biomass burning represent a significant fraction of absorbing aerosol globally, but it remains difficult to accurately predict SSA and AAE for biomass burning aerosol. Black carbon (BC), brown carbon (BrC), and non-absorbing coatings all make significant contributions to the absorption coefficient of biomass burning aerosol. SSA and AAE cannot be directly inferred based on fuel type because they depend strongly on burn conditions. It has been suggested that SSA can be effectively parameterized via the modified combustion efficiency (MCE) of a biomass-burning event and that this would be useful because emission factors for the MCE of a large number of fuels are available. Here we demonstrate, with data from the FLAME-4 experiment, that for a wide variety of globally relevant biomass fuels, over a range of combustion conditions, parameterizations of SSA and AAE based on the elemental carbon (EC) to organic carbon (OC) mass ratio are quantitatively superior to parameterizations based on MCE. We show that the EC/OC ratio and the ratio of EC/(EC+OC) both have significantly better correlations with SSA than MCE. Furthermore, the relationship of EC/(EC+OC) with SSA is linear. These improved parameterizations are significant because, similar to MCE, emission factors for EC (or black carbon) and OC are available for a wide range of biomass fuels. Fitting SSA with MCE yields correlation coefficients (Pearson's r) of ~0.65 at the visible wavelengths of 405, 532, and 660 nm while fitting SSA with EC/OC or EC/(EC+OC) yields a Pearson's r of 0.94-0.97 at these same wavelengths. The strong correlation coefficient at 405 nm (r = 0.97) suggests that parameterizations based on EC/OC or EC/(EC+OC) have good predictive capabilities even for fuels in which brown carbon absorption is significant. Notably, these parameterizations are effective for emissions from Indonesian peat, which have very little black carbon but significant brown carbon (SSA=0.99±0.07 at 532 and 660 nm, SSA = 0.93 ± 0.06 at 405 nm). Finally, we demonstrate that our parameterization based on EC/(EC+OC) accurately



predicts SSA during the first few hours of plume aging with data from Yokelson et al. (2009) gathered during a biomass burning event in the Yucatan Peninsula of Mexico.

## 1 Introduction

Black carbon (BC) aerosol is the dominant atmospheric absorber of visible light and has a significant impact on the Earth's radiative balance (IPCC AR5, 2013). It has been suggested that BC may have the largest positive radiative forcing after carbon dioxide (Jacobson, 2001; Bond et al., 2013). On a global scale, the largest source of BC is open burning of forests and savannas (Bond et al., 2013). Open biomass burning also contributes two-thirds of the primary organic aerosol (OA) emissions globally (Bond et al., 2004; Bond et al., 2013). Although most climate models treat organic carbon (OC) as purely scattering (Myhre G et al., 2007; Stier P et al., 2007), OC absorption can occur at shorter visible and ultraviolet wavelengths (Barnard et al, 2008; Lack et al., 2012a; Kirchstetter and Thatcher, 2012), which is commonly referred to as brown carbon (BrC). Few models account for BrC absorption or the effect of non-absorbing organic coatings increasing the absorption of BC (commonly called "lensing") in sophisticated ways (Jacobson, 2014; Lin et al., 2014). Recent measurements in the southeastern US show that biomass burning is the dominant source of brown carbon aerosol in this region (Washenfelder et al., 2015). Brown carbon can be a significant contributor to the overall aerosol absorption in biomass burning aerosol (Liu et al., 2014; McMeeking et al., 2014; Lack et al., 2012a). It has recently been suggested that the globally averaged top of atmosphere direct radiative forcing due to carbonaceous aerosols from biomass burning changes from negative to positive values when the effects of BrC are included (Saleh et al., 2015; Feng et al., 2013).

Including absorption from clear-coating enhancements and brown carbon in biomass burning aerosol into global models necessitates finding a simple and accurate parameterization of the optical properties for biomass burning aerosol. Direct microphysical modelling of aerosol absorption is not feasible because emissions, morphology, mixing state, and aging are not well understood and are not easily accessible from measurements. Even if such understanding and measurements were available, a microphysical model would be too computationally expensive. Single scattering albedo (SSA) and absorption angstrom exponent (AAE) are parameters that contain information on aerosol absorption and are commonly implemented in models and reported in observations. SSA and AAE are also critical for satellite retrievals (Ramanathan et al., 2001; McComseky, 2008) and uncertainty in SSA is one of the largest sources of uncertainty in estimating the aerosol direct and semi-direct effects (Jiang and Feingold, 2006; McComseky, 2008). SSA and AAE cannot be directly parameterized based on fuel type because they depend strongly on burn conditions and because there are no large datasets available that relate SSA or AAE to fuel type. It has been suggested that SSA and AAE can be parameterized based on the modified combustion efficiency (MCE), the ratio of $CO_2$ enhancement to the sum of CO and $CO_2$ enhancement (see Section 2.4), of a burn. Liu et al. (2014) parameterized SSA with MCE and accounted for much of the variation without including vegetation type into the parameterization, but the parameterization has limited predictive capability at higher MCE's (>0.92) where SSA changes rapidly (McMeeking et al., 2014). MCE has limited ability to predict aerosol BC/OA (Grieshop et al., 2009; Christian et al.,



2003) on which the absorptivity of organic aerosol has a strong dependence (Saleh et al., 2014). Recent studies show that utilizing the BC/OA mass ratio may be a better way to understand aerosol optical properties than via MCE (Lu et al., 2015; Saleh et al., 2014; McMeeking et al., 2014).

Here we present data collected during the Fourth Fire Laboratory at Missoula Experiment (FLAME-4). A wide range
of biomass fuels that represent significant sources of global biomass burning emissions were burned individually, and the resulting smoke was sampled with a range of in-situ instrumentation. SSA and AAE are parameterized with both the elemental to organic carbon mass ratio (EC/OC) and MCE showing that EC/OC is quantitatively superior. EC/OC is utilized in this study as opposed to BC/OA because of the experimental techniques employed (a Sunset Laboratories Instrument) though it is expected to yield similar results to BC/OA when parameterizations are applied to biomass burning emissions (Salako et al,
2012). Fuels studied include Indonesian peat, African grass, crop residue, US brushwood and coniferous trees. Indonesian peat is one of the largest sources of terrestrial organic carbon (Page et al., 2002). Previous FLAME studies were mainly constrained to the fuels prevalent in the United States (McMeeking et al., 2014; Lewis et al., 2008). Our study includes optical measurements at more wavelengths and covering a wider range of fuels than previous studies (McMeeking et al., 2014). Results are compared and contrasted with those of Liu et al. (2014), who also collected data during FLAME-4 with an
independent instrument suite, to demonstrate improvements in SSA and AAE parameterization with EC/OC versus MCE. The effectiveness our parameterization during the first few hours of aerosol aging is also demonstrated for a dataset from the Yucatan peninsula in Mexico (Yokelson et al., 2009).

## 2 Materials and Methods

### 2.1 Fourth Fire Lab at Missoula Experiment (FLAME-4)

Measurements were made during the FLAME-4 lab experiment, a multi-investigator experiment that took place from October 15 – November 16, 2012 at the Fire Sciences Laboratory in Missoula, MT. The combustion room at the Fire Sciences Laboratory measures 12.5 m x 12.5 m x 22 m high and has a 1.6 meter-diameter, 17-meter-tall exhaust stack with a 3.6 meter inverted funnel opening 2 meters above the fuel bed. The room was continuously pressurized with outside air that was conditioned for temperature and humidity. Details of the experimental setup and a summary of fuels burned during FLAME-
4 can be found in Stockwell et al. (2014). This paper summarizes results from 12 unique fuels and 41 individual burns. The list of fuels that were analysed and the geographic location where they were sampled from can be found in supplementary Table S1. Out of 41 individual burns, 20 burns were measured by pulling aerosol directly from the top of the exhaust stack (referred to as "stack" burns henceforth). During these stack burns, rapid variation in the aerosol properties are observed as the fire transitions from flaming to smoldering. The remaining 21 burns were measured from an inlet placed in the middle of
the combustion room (these are referred to as "room" burns henceforth). Sampling for the room burns began after the smoke was allowed to thoroughly mix in the combustion room (typically 15-20 minutes) and continued for several hours.



The suite of optical instruments used in this work was located in a room directly adjacent to the main combustion room. During both stack and room burns the smoke was transferred to the optical suite of instruments at 10 liters per minute via a ½" OD copper tube approximately 30 feet long. The transit time through the copper tube was roughly 5 seconds (and varied slightly depending on the amount of dilution air added for a particular burn). Smoke from the combustion room or stack was diluted to prevent signal saturation in the optical instruments. Dilution flow was generated from ambient air by passing it through a canister filled with Perma-Pure (Toms River, NJ) to remove $NO_x$ followed by a HEPA filter to remove particulates. Dilution air was introduced to the sample flow ~1 foot from the common inlet. All results presented in this paper explore intensive properties and thus are not sensitive to dilution unless significant evaporation of semi-volatiles occurred. While we cannot rule out evaporation of semi-volatiles from the particles, additional dilution of room burn particles is not expected to cause significant evaporation given that emissions were already diluted into the large combustion room. For stack burns, the additional dilution mimics the rapid dilution that would occur in the atmosphere. Given that different masses of fuel were burned during each individual burn (see SI Table 2 for details) the dilution factors for each burn already varied significantly because drastically different amounts of smoke were mixed into the combustion room or stack. The fact that our results show robust correlations over this wide range of dilutions strongly suggests that the exact amount of dilution was not a critical factor in parameterizing optical properties based on EC/OC.

## 2.2 Instrumentation

During the FLAME-4 experiment, absorption coefficients were measured by a 5-channel photo-acoustic absorption spectrometer (PAS) (Lack et al., 2012b) and extinction coefficients were measured by an 8-channel cavity ringdown spectrometer (CRDS) (Langridge et al., 2011). Absorption coefficients of dry aerosol (RH < 15%) were measured at 405, 532, and 660 nm and absorption of denuded aerosol were measured at 405 and 660 nm. The CRDS measured extinction coefficients of both dry and denuded aerosol at the same wavelengths as the PAS. In this paper, the denuded measurements are not used. The magnitude of absorption by the PAS was determined by sending ozone through both the CRDS and PAS and calibrating the PAS signal using the measured extinction from the CRDS, with Raleigh scattering subtracted (Lack et al., 2012b). The CRDS directly measures extinction without the need to calibrate (Langridge et al., 2011). Both the PAS and CRDS had a common inlet that conditioned the air through a number of steps. First, the aerosol was passed through a cyclone impactor that removed particles with aerodynamic diameters larger than 2.5 microns. Next, the air was dried by two 100-tube Nafion driers (Perma-Pure, Toms River, NJ) in parallel, which reduced the relative humidity in the sample cell to less than 15%. Following the Nafion driers, an activated carbon monolith (MAST Carbon, Basingstoke, UK) was used to scrub $NO_x$ and $O_3$ from the sample air while transmitting the particles. The removal of $NO_x$ was continuously tracked by a CRDS gas-phase channel at 405 nm. A filter was periodically inserted into the sample stream to remove particles and confirm baseline stability.

Uncertainty in the PAS absorption measurements is the sum of two terms, one accounting for calibration accuracy and the other for instrumental drift. The accuracy of the absolute calibration of the PAS is 5%, as detailed by Lack et al., (2006). The PAS was calibrated at the beginning and end of each measurement day during FLAME-4 and some instrument



drift was observed between calibrations. The maximum change in slope between the daily calibrations was 5%. Addition of these two 5% errors in quadrature yields the overall uncertainty for the PAS measurements (7%) for this study. For the CRDS, Langridge et al. (2011) demonstrated that the accuracy of measurements depends on relative humidity of the aerosol. For dry aerosol, as measured in this work, errors were shown to be < 2 % and that is the error utilized in this paper.

## 2.3 Calculation of SSA and AAE

Single scattering albedo is defined as:

$$SSA = \frac{b_{scat}}{b_{scat}+b_{abs}} = \frac{b_{scat}}{b_{ext}} \qquad (1)$$

where $b_{scat}$ is the scattering coefficient and $b_{abs}$ is the absorption coefficient. The sum of the scattering and absorption coefficients is known as the extinction coefficient ($b_{ext}$). SSA in this study was calculated via measurements of absorption by the PAS and extinction by the CRDS. Errors for SSA in this study were calculated by propagating the uncertainties described above for the extinction and absorption measurements and adding the standard deviation of the actual fire measurements in quadrature. Absorption angstrom exponent is defined as:

$$b_{abs} = a\,\lambda^{-AAE} \qquad (2)$$

where $b_{abs}$ is the absorption coefficient and the constant, a, is independent of wavelength. AAE is determined in this study from the slope of a least squares fit to the logarithm of absorption coefficients versus the logarithm of wavelengths. Data from three wavelengths (405, 532, and 660 nm) are used to determine AAE. Errors for AAE were calculated as one standard deviation of the slope of the least squares fit.

During the room burns, SSA and AAE were found to be nearly constant after emissions fully mixed in the dark combustion room and the results reported in this paper are the average of 1 Hertz measurements from an approximately 1-hour period after the smoke was fully mixed. The situation was different for stack burns, where rapid fluctuations in both SSA and AAE were observed as the amount of flaming and smoldering combustion within the fire varied. In order to obtain representative measurements for the stack burns, fire-integrated SSA and AAE were calculated. To generate fire-integrated SSA, the excess extinction and absorption coefficients were summed for the duration of the burn and then the SSA was calculated with these integrated parameters. This procedure gives more weight to periods of the burn that produced the most extinction or absorption and should generate numbers similar to what is observed during the room burns for burns with similar fractions of flaming and smoldering. Fire-integrated AAE is generated by summing the excess absorption coefficients measured during a stack burn then determining the AAE as described for the room burns.



**2.4 Modified Combustion Efficiency (MCE)**

The modified combustion efficiency is defined as

$$MCE = \frac{\Delta CO_2}{\Delta CO + \Delta CO_2} \tag{3}$$

where $\Delta CO$ and $\Delta CO_2$ are the background-subtracted CO and $CO_2$ mixing ratio (Ward and Radke, 1993; Yokelson et al., 1997). Background mixing ratios were measured before the ignition of each burn. The CO and $CO_2$ mixing ratio were measured by an open path Fourier transform infrared spectrometer (Stockwell et al., 2014). The MCE reported in this study is the fire-integrated value.

**2.5 Determination of the Elemental Carbon to Organic Carbon Ratio (EC/OC)**

Fine particulate matter ($PM_{2.5}$) was selected by a cyclone operating at a flow rate of 42 liters per minute and was collected on to 37 mm quartz fiber filters (QFF) (PALL, Port Washington, NY) at ambient temperature. Field blanks were collected at a rate of one in seven samples. Prior to use, QFF were pre-cleaned by baking at 550℃ for 18 hours. Filters were stored in cleaned aluminum foil-lined petri dishes sealed with Teflon tape, and stored frozen before and after analysis. OC and
15 EC were measured by thermal optical analysis (Sunset Laboratories, Forest Grove, OR, USA) following the IMPROVE-A protocol where the EC/OC split was determined by thermal optical transmittance. Analytical uncertainties for OC were propagated from the standard deviation of field blanks (0.7 µg cm$^{-2}$) and 5% of the OC concentration. For EC, uncertainties were propagated from an estimate of the instrument precision (0.1 µg m$^{-2}$), 5% of EC concentration and 5% of pyrolyzed carbon (which forms from OC charring on the filter during analysis). The value of 5 % is a conservative estimate of the
20 precision error in replicate sample analysis, which is typically 1-3 % (NIOSH., 1999). Analytical uncertainties for the EC/OC ratio were propagated from the individual EC and OC uncertainties.

**3 Results and Discussion**

Single scattering albedo (SSA) and absorption angstrom exponent (AAE) were measured during 41 individual burns of twelve different fuels during FLAME-4. Both quantities showed significant variability both from one fuel to another and
25 also between different burns of the same fuel.

**3.1 SSA Parameterization with MCE**

Figure 1 shows SSA and AAE plotted versus MCE. A consistent trend of decreasing values of SSA and AAE with increasing MCE is observed for all fuels. This trend has been observed before and can be explained by the fact that more BC and less OC is produced during the flaming part of a burn when MCE is highest, while more OC and less BC is produced



during the smoldering part of a burning when MCE is lowest (Ward et al., 1992; Christian et al., 2003; McMeeking et al., 2009, Liu et al., 2014). Accordingly, fuels that burn efficiently produce larger amounts of BC relative to OC and, because BC has significantly lower SSA than OC, the ensemble of particles from these efficient burns have lower values of SSA. Observed SSA values reached a minimum value of 0.26 ±0.02 at 660 nm. At high MCE, AAE is ~1 because BC absorption proportional to frequency. In contrast, fuels that burn with low efficiency are dominated by OC emissions, which predominantly scatter light resulting in SSA values nearing unity and larger values of AAE. The maximum AAE of 10.43 ±1.11 was observed for a peat burn. From the 9 different burns having MCE less than 0.90, the average SSA at 405 nm was 0.93 whereas the average SSA at 532 and 660 nm were 0.987 and 0.990 respectively. AAE values for these burns range from 3.68 ±0.47 to 10.43 ±1.11 with an average value of 6.51. These results strongly suggest that burns with low MCE have significant emissions of brown carbon, which has notable absorption at 405 nm, but is largely scattering at 532 and 660 nm.

While the general trend of decreasing SSA with increasing MCE is robust, a focus on the region with MCE greater or equal to 0.92 reveals that any parameterization based on MCE will have difficulty predicting the variability of SSA values in this region because SSA varies by up to 0.7 while MCE is nearly invariant. This difficulty has also been noted by McMeeking et al. (2014) based on data from the FLAME-3 set of experiments. Panels A and B of Fig. 1 show a least-squares fit to our FLAME-4 data along with the parameterizations proposed by Liu et al. (2014). The least-squares fits were made using the same functional form proposed by Liu et al. and allowing the coefficients to vary. Fitting coefficients and Pearson's correlation coefficients are given in Table 1. While the Liu et al. parameterizations have predictive capability, there are significant errors at high MCE suggesting that MCE may not be the optimal variable for parameterization. At 532 and 660 nm, there are also notable errors at low MCE. The fitted curves are significantly different than the data points at low MCE because the shape of the least-squares fitted curve is largely determined by the majority of points that occur at MCE > 0.90. A curve that fit the low MCE data better would have even larger errors than the curves shown at high MCE. The trends for AAE are similar to those for SSA. Panel D of Fig. 1 shows that parameterization of AAE with MCE can lead to significant errors at both low and high MCE.

Figure 2 demonstrates one reason why it is difficult to accurately predict SSA based on MCE. For similar values of MCE (at high MCE), EC/OC can vary by a factor of up to 6. This trend, which has also been noted by Christian et al. (2003) for a different suite of fuels, suggests that MCE is a poor proxy at the high MCE for the mass content of EC and OC on which absorption and scattering properties ultimately depend. Additionally, EC/OC for the same fuel was much more variable between lab burns than MCE. Eight different Sawgrass burns yielded MCE ranging in the narrow interval of 0.949 to 0.963, while EC/OC for the same burns varied from 0.010 to 4.358. Furthermore, this data demonstrates that EC/OC depends more on burn conditions than fuel type.

## 3.2 SSA Parameterization with EC/OC

Figure 3 shows the variation of SSA and AAE as a function of the EC/OC ratio for burns where EC/OC data from the Sunset Labs EC/OC instrument was available. At small EC/OC, aerosol composition is dominated by organic carbon




resulting in SSA values approaching unity (at wavelengths > 532 nm) and large AAE values (>2). In contrast, at large EC/OC, composition is dominated by elemental carbon, which produces minimum SSA values as low as 0.26 ± 0.02 at 660 nm and AAE values as low as 0.854 ±0.21. The wavelength dependence of SSA is pronounced when aerosol composition is dominated by organic carbon. In general, when there are significant amounts of organic carbon SSA values at 532 and 660 nm are fairly

close to one another while SSA values at 405 nm are significantly smaller. When composition is dominated by elemental carbon there is little dependence of SSA on wavelength and AAE values are less than 2.

These results are consistent with brown carbon causing absorption and lowering SSA at short wavelengths (<532 nm) (Barnard et al., 2008; Lack et al, 2012a; Kirchstetter and Thatcher, 2012) but only scattering at longer wavelengths. Figure 3 demonstrates that parameterization of SSA with EC/OC yields a function with a gentler change in slope that is less horizontal

at high SSA and will be shown to result in a lower errors in predicted SSA than parameterizations based on MCE. The least square fits of SSA to EC/OC have significantly larger correlation coefficients (average of 0.96 as shown in Table 2) than those for SSA fits based on MCE (average of 0.64 as shown in Table 1). The least squares fit of AAE to EC/OC gives a slightly larger r value (r = 0.80) than the fit of AAE to MCE (r = 0.74) but the improvement is less than for the SSA fits. The equations and Pearson's correlation coefficients for each fitting line are given in Table 2. All fits are constrained with an upper limit for

SSA of unity. Figure 4 demonstrates that when SSA and AAE are plotted vs. the EC/(EC+OC) ratio, the resulting fits are linear with similar correlation coefficients to the fits based on EC/OC.  The simple nature of the linear fit is appealing, but the data are more bunched because EC/(EC+OC) has significantly less dynamic range than EC/OC itself.  The y-intercepts of the fits vs. EC/(EC+OC), with SSA less than or near 1 at all wavelengths, are more physically realistic than the fits vs. EC/OC which significantly overshoot an SSA of 1 at low EC/OC ratios at 660 and 532 nm.

Given the effectiveness and linearity of the fit of SSA with EC/(EC+OC), one might conclude that the only truly important parameters for determining SSA are absorption by elemental carbon and scattering by organic carbon.  However, Figure 4 demonstrates that the y-intercept of the 405 nm fit is significantly less than the y-intercepts for the 532 and 660 nm fits. This suggests that the EC/(EC+OC) ratio is able to capture the effect of brown carbon, an idea that has also been suggested by Saleh et al. (2014; 2015).  In fact, the robustness of the fits suggests that the EC/(EC+OC) ratio is able to predict the SSA

and AAE despite the most likely complex and variable effects of particle size distribution, lensing, and brown carbon independent of the broad selection of fuel types tested, a rather surprising, but useful result.

To compare the performance of the different regression models discussed in this study, we calculated the root mean square error (RMSE) for the EC/OC and EC/(EC+OC) based models and compared with the RMSE of the MCE based model first proposed by Liu et al. (2014). The RMSE values for the different regression models are reported in Table 3. For the MCE

approach, the RMSE is similar whether coefficients from a least squares fit to our data are used or whether the coefficients proposed by Liu et al. (2014) are used. The main results is that the RMSE of the models based on EC/OC or EC/(EC+OC) are significantly less than RMSE for the model based on MCE.



### 3.3 Comparison of SSA and AAE from This Study with Previous Results

Given the useful robustness of the fits of SSA with EC/OC and EC/(EC+OC), it is important to confirm that the SSA values measured during this study are representative. As mentioned earlier, the FLAME-4 study analyzed a wide range of globally relevant fuels. Some fuels were burned in both FLAME-4 and FLAME-3 at similar MCE. In these cases the SSA
values from this study are, within experimental error, similar to those derived by McMeeking et al. (2014). The range of SSA values for ponderosa pine (varies from 0.83±0.06 to 0.98±0.07 ) and black spruce (varies from 0.90±0.06 to 0.93±0.06 ) at 532 nm are similar to previously reported SSA values measured on temperate and boreal forest at wavelengths of 540 and 550 nm (ranged from 0.83±0.11 to 0.97±0.02 ) (Hobbs et al., 1996; Radke et al., 1998; Radke et al., 1991). AAEs reported by Lewis et al., (2008) are comparable to AAEs estimated for burns with MCE greater than 0.90 during this study. However, for burns
with MCE less than 0.90, AAEs from this study are larger than many previously reported for biomass burning aerosols (Lewis et al., 2008; Liu et al., 2014), but McMeeking et al. (2014) is closer to our values. For burns with MCE less than 0.90, the AAE values from this study are comparable to previous studies that estimated the AAE of organic material, without BC, from biomass burning, as shown in Table 4. This is consistent with our study including burns (mainly peat combustion and two low efficiency ponderosa pine burns) that produced very little black carbon and behaved optically like pure organic emissions.
Overall, the data collected during this study are reasonably similar to previous studies, encompass a wide range of burn conditions, and produce new insights.

### 3.4 Impact of Parameterization with EC/OC

We next determined how significant an impact parameterizing SSA with EC/OC as opposed to MCE could have in climate models. Climate models typically begin with vegetation-based emission factors, so we compared the SSA predicted
by the MCE and EC/OC parameterizations using the literature average emission factors of a wide variety of global fires types compiled by Akagi et al. (2011). As shown in Table 5, the estimated SSA at 532 nm varies significantly for some fuels while it is similar for others. The differences for other wavelength is similar in magnitude and is given in Table S3. The maximum difference in the predicted SSA at 532 nm between EC/OC and MCE approach is ~ 48 % for garbage burning during. While climate models may not directly parameterize optical properties based on EC/OC, the parameterization provides a good sanity
check of model schemes to predict optical properties.

### 3.5 Robustness of the EC/OC parameterization of SSA in aging aerosol

Significant changes in the SSA of biomass burning aerosol can occur during the first few hours of aging (Abel et al., 2003; Yokelson et al., 2009; Vakkari et al., 2014). In a typical example, Yokelson et al. (2009) documented a BB plume where the SSA at 530 nm changed from ~0.75 to ~0.93 during 1.4 hours of aging. This presents a key question of whether the
parameterizations of SSA based on EC and OC presented in this work can capture the observed aging effects. To probe this question, we calculated the SSA that would be predicted based on our EC/(EC+OC) regression model and compared it to the



actual SSA observed by Yokelson et al. (2009) because that study provided reasonable auxiliary data. However, Yokelson et al. (2009) only measured $\Delta BC/\Delta PM_{2.5}$ instead of EC/(EC+OC) so several assumptions had to be made to implement our parameterization. $\Delta BC/\Delta PM_{2.5}$ was converted in to EC/(EC+OC) by choosing an average organic mass to organic carbon (OM/OC) ratio of 1.9 for the study (Yokelson et al., 2009), assuming all non-BC mass was OM, and assuming BC and EC

mass are identical. Figure 5 shows the measured and predicted SSA during the first 1.4 hrs of aging, which was the extent of aging observed in this dataset. The predicted SSA based on the EC/(EC+OC) parameterization closely tracks the measured SSA. The shaded region in Fig. 5 shows the 95% confidence interval of the predicted SSA values, and this interval encompasses most of the measured SSA values. This result suggests that the parameterization remains valid during initial aging of biomass plumes. A similar approach was used to check the performance of our SSA parameterization with observations from Vakkari

et al. (2014) of biomass burning plumes with variable ages in southern Africa. The predicted SSA at 660 nm shows reasonably good agreement with the measured SSA at 637 nm. The predictions are within roughly $\pm\,5\%$ for biomass burning plumes with SSA > 0.75, which encompasses most field observations of BB plumes. But the prediction do much worse for plumes with lower SSA. However, Vakkari et al. (2014) note that the "dark" plumes they observed are rarely observed in the atmosphere or laboratory studies.

## 3.4 Indonesian Peat

One fuel that deserves special mention is Indonesian Peat because of the large emissions from this source and the minimal available data for it. During the 1997 El Niño event, 0.19-0.23 Gt of carbon was estimated to be emitted to the atmosphere through peat burning, which was equivalent to ~40% of the mean annual global carbon emissions from fossil fuel (Page et al., 2002). Chakrabarty et al. (2015) recently presented results for emissions from Boreal peat, but not for Indonesian

peat, which is thought to currently be a more significant global source of emissions. Chand et al. (2005) reported an SSA of 0.99 for Indonesian peat burned in the laboratory at a wavelength of 540 nm. This SSA is similar to the SSA at a wavelength of 532 nm for Indonesian peat burned during this study ($0.99 \pm 0.07$) and by Liu et al. (2014). However, this study and Liu et al. (2014) present a significant new FLAME-4 finding that the SSA of the aerosol from burning Indonesian peat at 405 nm is $0.93 \pm 0.06$, demonstrating that this fuel emits significant amounts of brown carbon and is not only scattering. Our derived

average AAE of 7.7 further demonstrates the importance of brown carbon from this fuel and enables modelling of its absorption properties across the visible spectrum. Our AAE is higher than reported for the same peat fuels by Liu et al. (2014). Other peats (North Carolina, Canadian) produced aerosol with similar optical properties to Indonesian peat but currently burn less have less impact on the global radiative budget.

## 4.0 Conclusions

We examined the SSA and AAE of aerosol emissions from the combustion of a variety of globally significant biomass fuels under controlled laboratory conditions. It was found that SSA and AAE varied significantly, but that the variation could





be explained by dependence on burn conditions without a need to explicitly account for fuel type. Measured SSA ranged from $0.26 \pm 0.02$ to $1 \pm 0.07$ and AAE ranged from $0.85 \pm 0.21$ to $10.43 \pm 1.11$. We demonstrate that SSA parameterization with EC/OC and EC/(EC+OC) is quantitatively superior to parameterization with MCE and that the best fit of SSA with EC/(EC+OC) is linear. By applying various parameterizations of SSA to global emission factors measured by Akagi et al.

(2011), we demonstrate that parameterizations based on EC/OC and EC/(EC+OC) yield similar results to one another, but yield significantly different results than parameterizations based on MCE. Biomass burning aerosol emissions with compositions dominated by OC give higher AAE than those with more BC content, suggesting that the organic fraction of these emissions contains significant brown carbon. The effect of brown carbon on SSA parameterization with EC/(EC+OC) is to cause the y-intercept to be near unity (0.98-0.99) at 532 and 660 nm and less than unity (0.91) at 405 nm. Emissions from

the burning of peat, one of the largest source of terrestrial organic carbon (Page et al., 2002), yielded an SSA at 532 and 660 nm close to unity (0.99 on average), but an average SSA at 405 nm of 0.93. This wavelength dependent SSA, and an average AAE of 7.7, show that brown carbon absorption from peat combustion emissions are significant. Finally, we demonstrate that our parameterization of SSA based on EC/(EC+OC) accurately predicts SSA during the first few hours of plume aging with ambient data from Yokelson et al. (2009) gathered during a biomass burning event in the Yucatan Peninsula of Mexico.

*Acknowledgements.* This material is based upon work supported by the National Science Foundation under Grant No. 1241479. C. S. and R. Y. were supported primarily by NSF grant ATM-0936321. T. J. and E. S. were supported by University of Iowa. FSL operational costs were supported by NASA Earth Science Division Award NNX12AH17G to S. Kreidenweis, P. DeMott,

and G. McMeeking whose collaboration in organizing and executing FLAME-4 is gratefully acknowledged. We thank Ted Christian, Dorothy L. Fibiger, and Shunsuke Nakao for assistance with filter sample collection and sample preparation. We appreciate the contribution of Eric Miller, David Weise, Greg Askins, Guenter Engling, Savitri Garivait, Christian L'Orange, Benjamin Legendre, Brian Jenkins, Emily Lincoln, Navashni Govender, Chris Geron, and Kary Peterson for harvesting the fuels for this study. Collection of Indonesian peat by Kevin Ryan and Mark Cochrane was supported by NASA Earth Science

Division Award NX13AP46. We also thank Daniel Murphy for valuable suggestions during data collection manuscript preparation.



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





**Table 1.** Fitting coefficients for SSA as a function of MCE (SSA = $k_0$ + $k_1$(MCE)$^{k_2}$) and AAE as a function of MCE (AAE = a (MCE)$^b$) for fuels analyzed in this study with Pearson's r values for each fit. Numbers in parentheses are one standard deviation of the fitting coefficients.

|     | Wavelengths (nm) | $k_0$ | $k_1$ | $k_2$ | r |
|-----|-----|-----|-----|-----|-----|
| SSA | 405 | 0.920(± 0.043) | -0.632(± 0.221) | 26.877(±11.2) | 0.649 |
|     | 532 | 0.933(± 0.038) | -1.637(± 1.05) | 58.492(± 23.3) | 0.64 |
|     | 660 | 0.941(± 0.041) | -1.687(± 1.05) | 56.45(± 22.3) | 0.644 |
| AAE | 405/532/660 | a | b | | |
|     | | 2.454 (±0.231) | -3.292(±0.418) | | 0.738 |





**Table 2.** Fitting coefficients for SSA as a function of EC to OC ratio $(y = k_0 + k_1(EC/OC)^{k_2})$ and AAE as a function of EC to OC ratio $(y = a\,(EC/OC)^b)$ for fuels analyzed in this study with Pearson's r values for each fit. Numbers in parentheses are one standard deviation of the fitting coefficients.

| | Wavelengths (nm) | $k_0$ | $k_1$ | $k_2$ | r |
|---|---|---|---|---|---|
| SSA | 405 | 1.000($\pm$ 0.048) | -0.503($\pm$ 0.055) | 0.431($\pm$0.079) | 0.973 |
| | 532 | 1.000($\pm$ 0.043) | -0.476($\pm$ 0.059) | 0.578($\pm$ 0.119) | 0.960 |
| | 660 | 1.000($\pm$ 0.055) | -0.491($\pm$ 0.078) | 0.604($\pm$ 0.157) | 0.937 |
| AAE | 405/532/660 | a | b | | |
| | | 1.796 ($\pm$0.275) | -0.232($\pm$0.041) | | 0.796 |



**Table 3.** Root mean square error (RMSE) for SSA and AAE parameterization with EC/OC, EC/(EC+OC), MCE with fit coefficients from a least squares fit of this study's data, and MCE with coefficients from Liu et al., (2014).

|     | λ (nm) | EC/OC | EC/(EC+OC) | MCE (Liu et al., 2014) | MCE (Fit To This Study) |
| --- | --- | --- | --- | --- | --- |
| SSA | 405 | 0.041 | 0.04 | 0.163 | 0.115 |
|     | 532 | 0.057 | 0.043 | 0.17 | 0.133 |
|     | 660 | 0.077 | 0.06 | 0.224 | 0.144 |
| AAE | 405/532/660 | 0.911 | 0.92 | NA | 1.324 |



**Table 4.** AAE calculated for burns with MCE less than 0.90 compared with previous studies that derived the AAE of organic mass (without BC) for biomass burning emissions.

| Reference | Wavelength (nm) | Aerosol Component Analyzed | AAE |
|---|---|---|---|
| This Study | 405/532/660 | Entire Aerosol | 3.7 to 10.4 Average = 6.5 |
| Srinivas, B. and Sarin, M. M. (2013) | 365-700 | Water Soluble Organics | 3 to 19 Average = 9 |
| Srinivas, B. and Sarin, M. M. (2014) | 300-700 | Water Soluble Organics | 8.3 ± 2.6 |
| Feng et al., (2013) | 400-700 | Organic | 6.6 |
| Sun et al., 2007 | NA | Water Soluble Organics | 6 |
| Zhong, M. and Jang, M (2014) | 400-700 | Organics | 4.74 |
| Kirchstetter and Thatcher (2012) | 400-700 | Organics | 3.0 to 7.4 Average = 5 |




**Table 5**. Comparison of SSA at 532 nm predicted based on MCE (with Liu et al. (2014) coefficients), EC/OC, and EC/(EC+OC) parameterizations for emission factors of various biomass fuels from Akagi et al. (2011). The values in % difference in predicted SSA column are percentage difference for EC/OC / EC/(EC+OC) approaches respectively. BC/OC is assumed to be equivalent to EC/OC.

| Biomass Types | MCE[1] | BC/OC[1] | SSA_532 MCE Approach | SSA_532 EC/OC Approach | SSA_532 EC/(EC+OC) Approach | % Difference In Predicted SSA |
|---|---|---|---|---|---|---|
| Tropical Forest | 0.95 | 0.11 | 0.79 | 0.87 | 0.88 | -10.13 /-11.39 |
| Savanna | 0.96 | 0.14 | 0.67 | 0.85 | 0.86 | -26.87 /-28.36 |
| Crop Residue | 0.94 | 0.33 | 0.82 | 0.75 | 0.73 | 8.54/10.98 |
| Pasture Maintenance | 0.92 | 0.09 | 0.89 | 0.88 | 0.89 | 1.12 /0 |
| Boreal Forest | 0.92 | - | 0.88 | - | - | - |
| Temperate Forest | 0.95 | - | 0.77 | - | - | - |
| Extra tropical Forest | 0.93 | 0.07 | 0.87 | 0.9 | 0.92 | -3.45/ -5.75 |
| Peat land | 0.9 | 0.03 | 0.94 | 0.93 | 0.95 | 1.06 /1.06 |
| Chaparral | 0.96 | 0.35 | 0.69 | 0.74 | 0.72 | -7.25 /-4.35 |
| Open Cooking | 0.95 | 0.29 | 0.75 | 0.77 | 0.76 | -2.67/-1.33 |
| Patsari Stoves | 0.97 | 0.39 | 0.58 | 0.73 | 0.7 | -25.86 /-20.69 |
| Charcoal Making | 0.86 | 0.03 | 0.97 | 0.94 | 0.96 | 3.09 /1.03 |
| Charcoal Burning | 0.93 | 0.77 | 0.87 | 0.59 | 0.54 | 32.18 /37.93 |
| Dung Burning | 0.89 | 0.29 | 0.95 | 0.77 | 0.75 | 18.95 /21.05 |
| Garbage Burning | 0.97 | 0.12 | 0.58 | 0.86 | 0.87 | -48.28 /50 |

[1]Data from Akagi et al., 2011





**Figure 1.** Single scattering albedo at (a) 405 nm, (b) 532 nm, and (c) 660 nm plotted as a function of MCE measured during the FLAME-4 experiment. The solid red lines are least-squares fits of the data with details given in Table 1 and the black lines are parameterizations proposed by Liu et al. (2014) which have the same functional form as the red-line fits. Error bars for SSA are calculated by taking 7% uncertainty in absorption measurement, 2% uncertainty in extinction measurement and one standard deviation of the average value for room burns and adding these uncertainties in quadrature. For stack burns, the error is the quadrature sum of the uncertainty in the PAS and CRDS measurements. (d) Absorption angstrom exponent as a function of MCE. The error bars for AAE are one standard deviation of the least squares fit to the averaged data for a given fuel.



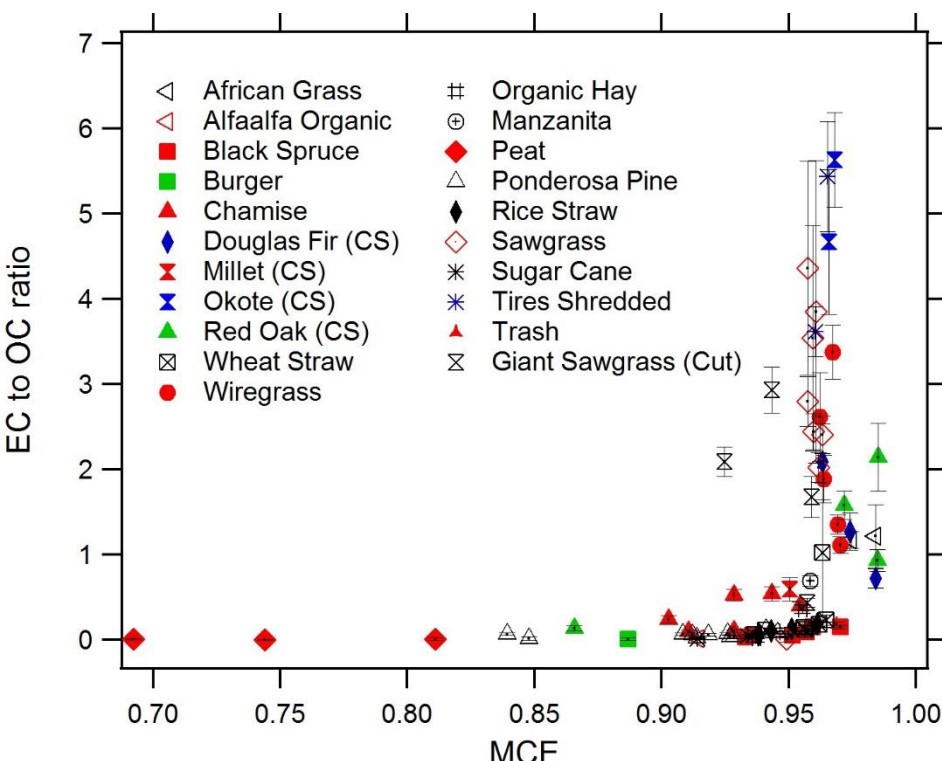

**Figure 2.** EC/OC ratio plotted as a function of MCE. Different symbols and colours represent different fuels as listed in the legend. The error bars are the propagation error calculated from the uncertainty associated with the EC and OC measurement. This plot contains data from additional fuels from FLAME-4 for which SSA and AAE were not calculated.





**Figure 3.** SSA at (a) 405 nm, (b) 532 nm, and (c) 660 nm plotted as a function of EC/OC. Solid red lines are least-squares best fits of the data that are constrained to force SSA to be less than one. The error bars on SSA are the propagation error calculated by taking 7 % uncertainty in absorption measurement, 2% uncertainty in extinction measurement and one standard deviation of the average value for room burns. Whereas for stack burn it is the propagation error calculated form uncertainty in PAS and CRDS measurements. Error bars on the EC to OC ratio are the propagation error calculated from the uncertainty associated with the EC and OC measurement (d) AAE as a function of EC/OC with error bars being one standard deviation of the measurement.





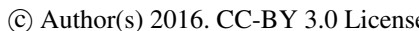

Figure 4. Same as Fig. 3 but for SSA plotted vs. $\frac{EC}{(EC+OC)}$. Equations for the fits are given in each panel.





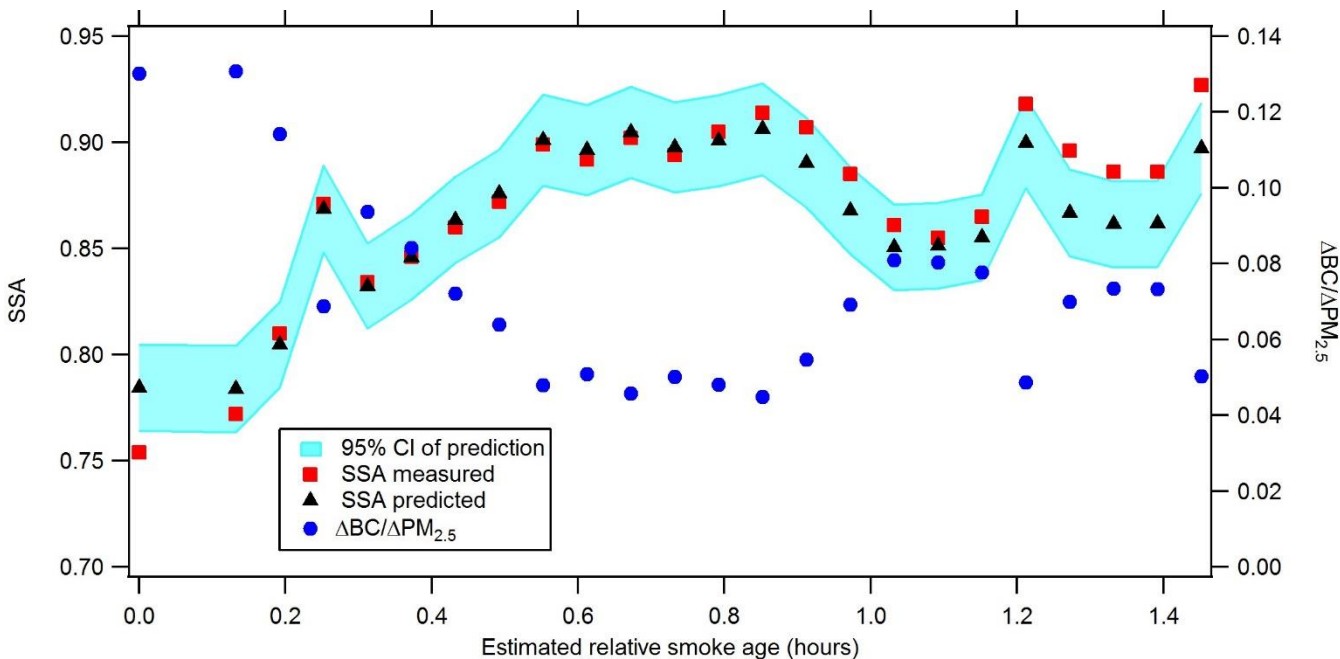

**Figure 5.** Measured (Yokelson et al., 2009) and predicted SSA at 532 nm for biomass burning emissions during the initial 1.4 hours of aging. Shaded region indicate the 95 % confidence interval of the predicted SSA.

