# Peer review of "Parameterization of Single Scattering Albedo (SSA) and Absorption Angstrom Exponent (AAE) with EC/OC for Aerosol Emissions from Biomass Burning"

_Atmospheric Chemistry and Physics, 2016_

## Referee Comment (RC1) · Anonymous Referee #3 · 15 Apr 2016

This paper presents results from the FLAME IV measurement campaign, in which a range of biomass types, the emissions from which are of great atmospheric relevance, were burned under controlled conditions and monitored by a number of groups. This paper describes measurements of the optical properties of emitted particles (absorption and scattering at multiple wavelengths) along with complementary measurements of organic and elemental carbon (OC/EC), CO and CO2 during these burns. Optical measurements yield estimates for aerosol single scattering albedo (SSA) and absorption angstrom exponent (AAE), which are commonly used in atmospheric models and satellite data retrievals. Previous efforts have shown that the modified combustion ef-

ficiency of a combustion event can be linked to the SSA of the emitted particles. The paper proposes a different relationship that relates SSA and AAE to the EC/OC or EC/(OC+EC) ratios and show that this relationship has better predictive power than that based on MCE over a wider range of conditions. In particular, the MCE-based parameterizations tend to do a poor job at high MCEs (>0.92), which are characteristic of more complete, typically flaming-phase combustion, and also potentially at low MCEs, as fits are distorted due to the prevalence of higher MCE conditions. The OC/EC based parameterizations result in improved fits of the optical properties of particles emitted during combustion of the wide range of fuels. Based on this, the authors suggest that these parameterizations can improve modelers' abilities to represent particle optical properties for both fresh and potentially aged emissions. To support the latter, they include an application of the method to data collected during measurements within a large biomass burning plume, which suggest that the parameterization can be used to track optical properties in an evolving plume.

Improving our understanding of the optical properties of biomass burning emissions is an important goal, and this paper makes a substantial contribution to that, and suggests a way in which available data can be leveraged to improve the representation of particles in models and satellite retrievals. While the work is suited for eventual publication in ACP, there are a number of issues that should be more comprehensively addressed before it is published. In general, the paper is well-written and clear, and should be a valuable contribution to ACP once these issues are addressed.

Major points:

A key concern is with the manner in which OC was quantified, and the potential influence of measurement artifacts on its determination. While the authors offer some discussion of the potential influence of gas-particle partitioning on the observed OC levels, they make no mention of the influence of gas-phase artifacts on quartz-fiber filters, which were used for determination of OC. It has long been known that positive artifacts can contribute the majority of OC mass in some cases (Subramanian et al.

2004). The fact that OC measurements were collected on bare quartz filters, means that there will be a substantial gas-phase artifact, which likely has a larger high-biasing effect on filters with lower OC levels. It is well established at this point that OC from biomass combustion has a large contribution from semivolatile materials, that will partition based on environmental conditions and also readily lead to positive sampling artifacts on quartz filters (Grieshop et al. 2009; Lipsky and Robinson 2006; May et al. 2013). While it is not possible for the extent of this artifact to be measured now (assuming no other groups have done so) an estimate for this effect could be determined using the volatility parametrization from May et al, collected during Flame III. This would also enable a quantitative examination of the impact of dilution on comparison between burns, which was dismissed as unimportant in a not-convincing way (P4, L7-13). Levels of dilution can have substantial effects on the partitioning of organics (See e.g. Fig. 6 in (May et al. 2013)) and since your parameterization is a direct function of OC, it is important to eliminate any biases in the measurements. Overall the authors should be applauded for careful uncertainty analysis, which is too often ignored, but the influence of these processes/artifacts should be included in this analysis. Such artifacts present a complication to the application of this kind of parameterization because both OC and EC measurements are operationally defined and sensitive to sampling conditions. Sampling conditions (average sample concentrations) should be included in the supplement, and efforts made to remove biases resulting from different sampling conditions.

A general comment on the paper is that a number of real-time properties of emissions were characterized at 1 Hz, but only 'burn-average' properties discussed. It would be interesting to see how these properties evolved for individual burns as it progressed, as in many cases there will be distinct phases with different properties, and the relative prevalence of these different properties may be very different for the same fuels in different conditions.

It would be helpful if confidence intervals were provided on regression fits.

[Figure]

One important detail left out of the final section, comparing results from the (Yokelson et al. 2009) study, is that 'PM2.5' and 'BC' in this study were both determined optically (via nephalometer and PSAP, respectively). It is therefore a bit circular to use these to show that a parameterization based on chemical measurements can be used to represent optical properties. The 'calibration' of the PM/BC measurements in Yokelson et al does provide an indirect link to 'mass' measurements, but they are still optical measurements. This still may be a useful example of the applicability of your result, but needs to be used with proper caveats. To properly do uncertainty analysis on this, the uncertainty in the other assumptions (MAE, MSE) used to estimate BC and PM should also be included.

Minor Points

P3, L10-12 – This sentence is confusing. It makes it sound as if Indonesian Peat is the largest source of organic carbon on the ground (terrestrial). Also, combustion of peat is a varying source and I don't know if a statement so strong is justified. P4, L6 – Pretty sure you can't put Perma-pure in a canister this way? Did you use a Nafion dryer? Maybe thinking of some other compound? P4, L13-15 – This is a circular argument unless you have some a priori reason that the quantities you are comparing should have robust correlations. P5, L27 – clarify what is meant by excess P7, L4 – missing a word here, perhaps 'is'? P8, L8-9 – It is not stated what the chosen functional form is, and why it was chosen. P8, L20 – much of this strong correlation is driven by the fact that there are two clusters of data that are widely spread, through which a line can be drawn. An exponential-type curve could also be driven, and might asymptote at a more reasonable value as EC/TC goes to higher values. P8, L24-26 – This sentence is awkward and difficult to understand. P 9, L15 – it would be helpful if some of this comparison were made graphically, either as a separate plot, or by adding some/all of these points on existing plots P9, L21-22 – Is this really all that can be said about this? If this is the case, it's really not clear whether it is worth including a table, especially in the main paper. If no systematic point can be made, put the table in the SI and just

include a range of differences. P9, L23 – Extra word? P9, L24-25 – Worth a discussion if you are saying your proposed parameterizations won't be applied in models. This is your original motivation – why would it only be used in this limited way if it is so much better than the alternatives? P10, L12-13 – sentence fragment. This should be quantified: 'much worse'. EC-dominated combustion is more common in biofuel use, so may still be an issue. P10, L26 – Any suggestions as to why the AAEs determined for the same burns are so different? P10, L27-28 – There should be a reference to a source with these data. Also, seems to be a missing word in this sentence. P11, L9 – Would be good to mention the physical significance of this intercept. Table 5 –should be no more than 2 significant figures in % difference column

References

Grieshop, A. P., Miracolo, M. A., Donahue, N. M., and Robinson, A. L. (2009). "Constraining the Volatility Distribution and Gas-Particle Partitioning of Combustion Aerosols Using Isothermal Dilution and Thermodenuder Measurements." Environmental Science & Technology, 43(13), 4750–4756.

Lipsky, E. M., and Robinson, A. L. (2006). "Effects of dilution on fine particle mass and partitioning of semivolatile organics in diesel exhaust and wood smoke." Environmental Science & Technology, 40(1), 155–162.

May, A. A., Levin, E. J. T., Hennigan, C. J., Riipinen, I., Lee, T., Collett, J. L., Jimenez, J. L., Kreidenweis, S. M., and Robinson, A. L. (2013). "Gas-particle partitioning of primary organic aerosol emissions: 3. Biomass burning." Journal of Geophysical Research: Atmospheres, 118(19), 2013JD020286.

Subramanian, R., Khlystov, A. Y., Cabada, J. C., and Robinson, A. L. (2004). "Positive and negative artifacts in particulate organic carbon measurements with denuded and undenuded sampler configurations." Aerosol Science and Technology, 38, 27–48.

Yokelson, R. J., Crounse, J. D., DeCarlo, P. F., Karl, T., Urbanski, S., Atlas, E., Campos,

T., Shinozuka, Y., Kapustin, V., Clarke, A. D., Weinheimer, A., Knapp, D. J., Montzka, D. D., Holloway, J., Weibring, P., Flocke, F., Zheng, W., Toohey, D., Wennberg, P. O., Wiedinmyer, C., Mauldin, L., Fried, A., Richter, D., Walega, J., Jimenez, J. L., Adachi, K., Buseck, P. R., Hall, S. R., and Shetter, R. (2009). "Emissions from biomass burning in the Yucatan." Atmos. Chem. Phys., 9(15), 5785–5812.

---

## Referee Comment (RC2) · Anonymous Referee #1 · 16 Apr 2016

The paper by Pokhrel et al. summarizes measurements of optical properties, namely SSA at 405 nm, 532, and 660 nm and the angstrom exponent of absorption, of biomass burning aerosol formed in the lab. They provide parameterization of these properties as a function of EC/OC or EC/(OC+EC) and demonstrate how such parameterizations are more robust compared to the parameterizations against modified combustion efficiency (MCE) for a variety of biomass fuels and burn conditions. The concept that brown carbon concentration emitted in fires can be related to BC/OA has already been put forth, and so it's not surprising to have correlations between SSA and EC/OC since these parameters are all related. The uniqueness of the paper is in determining the

[Figure]

SSA/AAE for the different samples and to determine the new parameterizations that are more robust when EC and OC are readily available. I support publishing the manuscript after the following comments are addressed.

Holistic comments:

Which parameter is more easily and accurately available for fires on regional and global scales? MCE or the amount of BC and BB OC (or OA)? Since OA/CO ratios for fires are quite variable and hard to predict in models, which SSA parameterization would then lead to the least uncertainty in the radiative effects of wildfire aerosols? See also my specific comment below.

MCE, EC/OC, and EC/(OC+EC) all have uncertainties/errors associated with them, so the fits are more appropriate if they're ODR with uncertainties as weights of the fit and not least-square linear regression lines as is done throughout the paper. (This is the main reason for my rating of the paper as 'major' revisions since all the figures, tables, and numbers in the text are to be updated accordingly).

Other specific comments: P4, L8: The authors discuss possible evaporation of semi-volatile components after sample dilution, but another factor is temperature differences in the relatively long sampling line. What was the temperature of the sampling lines kept at? If it wasn't controlled, how does this temperature difference impact redistribution of semi-volatile components of aerosols?

P4, L28: How is the response of activated carbon monolith to gaseous organic species? Could this sample treatment introduce negative artifacts in organic aerosols?

P5, L27: What does 'excess' extinction and absorption mean? I'm not sure how SSA/AAE for stack burns in calculated- I thought it should be just based on average values of abs and ext, but not sure what summing up the 'excess' amounts mean.

P6, Section 3.1: I agree that the data suggest EC/OC ratios are more variable for a given fuel than MCE and therefore relationships of optical properties with EC/OC

are more robust. Can the authors elaborate on what specifically changes in the 'burn condition' that leads to such variability in aerosol characteristic? Is the different in the water content of the fuel or the starting temperature of the fire, etc. etc?

P9, Section 3.4: Can you perform a simple calculation to estimate instantaneous TOA forcing difference of BB aerosol depending on the choice of SSA (new vs. old parameterization) so readers get an idea about the magnitude of the change in forcing? This calculation should be done considering the uncertainties in MCE, EC/OC (or EC/(OA+EC)) and SSA.

P9, L19: I'm confused by the statement "While climate models may not directly parameterize optical properties based on EC/OC, the parameterization provides a good sanity check of model schemes to predict optical properties." If models have EC/OC data to do this sanity check, do the authors not recommend the modelers to use this parameterization instead of other estimates? If yes, I think the sentence needs to be rephrased. If not, elaborate why this shouldn't be the recommended approach.

P10, line 16-18: I do understand that the average SSA of peat burning aerosol is lower, but given the uncertainty for the SSA values, the difference at 532 nm vs. 405 nm is not really significant. The relatively high AAE is more convincing for the presence of BrC in peat burning. What is the uncertainty in AAE for this sample? Add that value as well.

P10, L 30: I question the assumption of PM2.5 in a fire being composed of only BC and OM. In most fires, there could also be aerosol nitrate and chloride. How does the ratio of EC/(EC+OC), and therefore, estimate of SSA change if say 5-15% of PM2.5 is assumed to be inorganics? Also, looking at Table 3 in Yokelson et al., ACP 2009, there were direct PM1-OM measurements. Why not use that measure of OM when calculating OC?

Additionally, BC and EC are not necessarily presenting the same type of species. Can you reference papers that perform both measurements on a series of burned fuels and

comment on the 'goodness' of this assumption and how it will impact the predicted SSA?

Minor comments:

Page 2, L7: primary OA=POA

Page 3, L16: 'effectiveness of our . . ."

P7, L 3-4: "At high MCE, AAE is ∼1 because BC absorption proportional to frequency" incomplete sentence. Also, by frequency, do you mean 'wavelength'?

P7, L4-5: should clarify that low MCE burns give high SSA at long wavelengths, since in the following sentences it's mentioned that the OA in low MCE burns is highly absorbing as BrC.

P9, L 13: consider ". . ..how significant of an impact. . ."

P9, L18: during . . .? Incomplete sentence

P 10. Indonesian Peak section should be 3.6

P10, L6: What's the explanation for the parameterization not capturing the measured SSA in plumes with lower SSA? Also, tart the sentence with "However" instead of "But", or combine the two sentences

P10, line 21-22: Rephrase the sentence, sounds like summary bullets and not a complete sentence.

P11, line 27: When giving the range of SSA, separate it for 405 nm vs. 660 nm.

P12, Line 5-6: Indicate again the errors bars for SSA of peat aerosol at 405 nm and longer wavelengths as well as the error bar for AAE.

---

## Referee Comment (RC3) · Anonymous Referee #4 · 19 Apr 2016

Review of
**"Parameterization of Single Scattering Albedo (SSA) and Absorption Angstrom Exponent (AAE) with EC/OC for Aerosol Emissions from Biomass Burning"**
By Pokhrel et al.

General comments

This paper demonstrates a novel approach in parameterizing SSA and AAE for aerosol emitted from biomass burning using EC/OC ratio instead MCE. The parameterization using EC/OC and EC/(EC+OC) ratios were based on burning experiments of different kind of fuels, including Indonesia peat. The findings show that these parameterizations have better prediction for fuel with significant loadings of black carbon and/or brown carbon. Additionally, the parameterizations were able to predict SSA from biomass burning plume aging in Mexico.

Generally, the manuscript is well written. There are only several typos and some statements that need clarification (see technical comments). The authors carefully estimate the uncertainties of their methods and parameterizations. I only have couple questions that may need to be clarified and some information may need to be added.

Overall, I recommend publishing this manuscript after the questions are addressed.

Specific comments
1) Pg 3 Ln 25-26. The authors explain about observation of fire transitions from flaming to smoldering. It would be good to add information or clarify how was this done. Also how fractions of flaming to smoldering (Pg 5 Ln 28-30) were maintained during different experiments.
2) Pg 7 Ln 29. This statement about EC/OC depends more on burn conditions than fuel type can mislead. As it is shown in this study (Fig. 2), for certain fuels, such as peat, the different combustion types do not change EC/OC significantly. Therefore, fuel types do influence the EC/OC ratio.
3) Pg 8 Ln 24-26. The authors suggest that EC/(EC+OC) ration is able to predict AAE. I think the r of the least square fit is not that strong for this case (r=-0.79) different from fitting for SSA. The statement may need to be revised.
4) Pg 8 Ln 29-30. I think the RMSE values from fit of this study and fit by Liu et al. (Table 3) are not similar. They are about 20-30% different. The statement may need to be revised.

Technical comments
1) Pg 4 Ln 6. What does it mean "a canister filled with Perma-Pure"? Perma-Pure is a manufacturer name.
2) Pg 6 Ln 14. What is temperature of filter storage?
3) Pg 7 Ln 18-19. I am confused with this statement. Liu et al. provided parameterization only for 405 and 532 nm, and on Fig. 1, there is no black fitted line for panel C (660 nm). So what does 660 nm refer to?
4) Pg 9 Ln 22-23. This sentence is not finished?
5) Pg 10 Ln 10. Provide correlation plot and value of SSA at 660 nm and at 637 nm in SI.
6) Pg 10 Ln 22-25. In which table we can find the SSA and AAE for Indonesian peat and the other fuels?

---

## Referee Comment (RC4) · Anonymous Referee #2 · 11 May 2016

The authors present measurements of aerosol optical properties for particles produced from biomass combustion. The confirm that there is a relationship between the SSA and AAE and the MCE, as has been shown before. But they importantly extend this to think about the relationship with the EC/OC ratio (and the EC/(EC+OC) ratio), finding a stronger relationship with these variables. These results may be useful for global modeling (although it should be noted that global models tend to calculate optical properties based on particle size distribution and composition and do not simply specify SSA as

an a priori parameter). I think that this paper should be publishable after the below comments are addressed. My main question at this point relates to their choice of r as the figure of merit in their (non-linear) fitting.

P1,L17: Suggest replacing "significant" with "substantial" or "important" so as not to imply statistical significance.

P1,L18: Suggest replacing "inferred" with "predicted".

P1,L20: I find ". . .emission factors for the MCE. . ." to be unclear. EFs of what?

P1,L27 and General Question: Pearson's r is a parameter that describes the linear correlation between two variables. Here, it seems to be applied to one data set that is linearly related (SSA vs. EC/(EC+OC)) and two that are not (SSA vs. MCE and SSA vs. EC/OC). Thus, are the r values really comparable? What does an r value mean for a non-linear relationship? Might a different statistical test be applied? Perhaps Spearman's or Kendall's rank correlation coefficients or a Pearson's Chi Square test?

P2,L9: The authors cite Stier (2007) as evidence that "most climate models treat organic carbon as purely scattering." However, it is clear from Fig. 1 in Stier (2007) that the OC is somewhat absorbing throughout the visible. In fact, most models treat OC as slightly absorbing (see e.g. the OPAC database).

P2,L13: The inclusion of the reference to Washenfelder here seems quite selective, as there are many regions where biomass burning has been implicated as a source of BrC. Not that it is not a nice study, but is there a reason why this study is being highlighted?

P2,L17: I suggest that the Saleh reference is removed and only the Feng (global model result) reference is retained.

P2,L23: What is meant by SSA and AAE are "commonly implemented in models"? Models don't specify SSA. Similarly, what is meant by "SSA and AAE are also critical for satellite retrievals"? Critical for or are important retrieved information from?

[Figure]

P3,L9: No reference to Salako et al. is provided. Also, I would contend that this really remains to be demonstrated as "charring" is known to be a particularly important for biomass burning. Also, the authors might compare their longest wavelength denuded-particle absorption measurements to the EC measurements to argue that there is a reasonable relationship between BC and EC for this data set.

P7,L3: The sentence starting "At high MCE" is a fragment.

Fig. 1: It seems odd that the least squares fit (red line) doesn't match the data at smaller MCE values at 660 nm (most notably). Is there a reason for this? The functional form used (which should be given in the main text as well) should allow for better agreement at these low MCE values. Also, it is unclear if the fits were performed with/without accounting for the uncertainties in the individual points.

Fig. 3: The fits the authors retrieve allow for unphysical SSA values > 1. I suggest that they redo their fits, constraining the maximum retrievable SSA to be <= 1. This amounts to constraining the k0 in their fit equation to be <= 1. This links to P8,L14, where the authors note that this fitting does lead to SSA values > 1. But this is a solvable problem. Physical realism can be imposed on the fits.

P8,L1: the authors might indicate what they consider the EC/OC value at which composition is "dominated" by EC.

P8,L5: horizontal should be vertical.

P8,L12: Are the data truly more "bunched" or is the difference that Fig. 4 uses a linear scale and Fig. 3 a log scale for the x-axis? I think the latter.

P8,General: The authors discuss the robustness of their fits and the ability of EC/(EC+OC) to be used as a predictor. Although I generally agree, a few thoughts: (i) I think that the authors are overstating the case for AAE, as the correlation coefficient is only 0.79. (ii) Regarding the 405 nm measurements, yes, the fit gets a <1 SSA value when EC/(EC+OC) = 0. But it is also clear that the zero intercept here differs

substantially from the data points. In other words, the fit is certainly "good" but the model fit and observed SSA values differ by ~0.03 or more, which is small yet non negligible. (iii) Can the authors include confidence bands?

Table 5 and discussion: Do the MCE and EC/OC from the literature for biomass burning emissions generally agree with the observations here in terms of functional form?

P9,L19: I find the meaning of the following sentence to be unclear: "While climate models may not directly parameterize optical properties based on EC/OC, the parameterization provides a good sanity check of model schemes to predict optical properties." Can the authors clarify how this table and discussion provides a "sanity check"?

P10,L4: What is meant by "reasonably good?" As good as the case that is shown? Can this just be shown?

P10,L12: If the peat burning was unintentional and a result of e.g. drought, I suggest the authors say "through unintentional peat burning."

P7,L5: To set things up for later in the paper, the authors might report the mean value for peat here in addition to the maximum. Some discussion of the variability would also be helpful (later in section 3.4).

P10,L21: this is a sentence fragment.

General: I suggest that the authors adopt the terminology "aerosol particles" throughout much of the particle, to indicate that they are looking at the particulate matter and not the associated gaseous material.

---

## Author Comment (AC1) · 25 Jun 2016

**We would like to thank the reviewer for their valuable suggestions and time. Our responses are given below.**

**Anonymous Referee #3**

**Referee Comment:** A key concern is with the manner in which OC was quantified, and the potential influence of measurement artifacts on its determination. While the authors offer some discussion of the potential influence of gas-particle partitioning on the observed OC levels, they make no mention of the influence of gas-phase artifacts on quartz-fiber filters, which were used for determination of OC.

**Author Response:** We agree with the reviewer that gas sorption onto quartz fiber filters (QFF) can introduce positive sampling artifacts. To assess such artifacts, back-up quartz fiber filters were collected behind Teflon filters during sample collection in FLAME-4. The quartz filters behind Teflon (QBT) adsorb gases only and thus serve as a measure of gases adsorbed on QFF during sampling (Cheng et al., 2009). For fourteen FLAME-4 burns, representing five different biomass types (grass, rice straw, pine, spruce, peat) gas sorption accounted for an average of 2.4 ± 1.2 % of the OC (ranging from below the instrument limit of detection to 4.7%). In light of the reviewer's comment, we have applied the artifact correction to this data set and describe this in the methods section of the text. All figures and data analysis in the revised manuscript are based upon artifact-correction data.

**Added Text Location:** Section 2.5, page 6, line 16

**Added Text:** The effects of positive sampling artifacts due to carbonaceous gas adsorption were assessed using quartz filters behind Teflon (QBT) (Cheng et al., 2009) for 14 of the 96 fires, including grass, rice straw, ponderosa pine, black spruce and peat. For fires with QBT collected, the OC on the backup filter was subtracted directly. For fires without backup filters or those that were below the detection limit, the average OC correction for that fuel type was applied: rice straw (2.0 ± 0.4 %), ponderosa pine (1.2 %), black spruce (2.9 ± 1.6 %) and peat (3.1 ± 0.8 %). For fuels types without backup filters collected, the study average OC artifact (2.4 ± 1.2 %) was subtracted.

**Referee Comment:** …using the volatility parametrization from May et al, collected during Flame III. This would also enable a quantitative examination of the impact of dilution on comparison between burns, which was dismissed as unimportant in a not-convincing way (P4, L7-13). Lev-els of dilution can have substantial effects on the partitioning of organics (See e.g. Fig.6 in (May et al. 2013)) and since your parameterization is a direct function of OC, it is important to eliminate any biases in the measurements.

**Author Response:** We agree that the effect of dilution can be significant and have modified the text on P4, L7 to more explicitly address this issue.

**Added Text Location:** Section 2.1, page 4, line 4

**Added Text:** Smoke from the combustion room or stack intended for analysis by the optical

suite of the PAS and CRDS was diluted to achieve extinctions of approximately 500 Mm$^{-1}$ or less to prevent signal saturation in the CRDS. Dilution flow was generated from ambient air by passing it through an active-charcoal and permanganate (Purafil) scrubber to remove gas phase absorbers ($O_3$ and $NO_x$) followed by a HEPA filter to remove particulates.  Dilution air was introduced to the sample flow ~1 foot from the common inlet.  All results presented in this paper explore intensive properties and thus are not sensitive to dilution unless significant evaporation of semi-volatiles occurred, which has been shown to be possible (May et al., 2013). All emissions, weather additionally diluted for optical measurements or not, experienced significant dilution before sampling.  A wide range of dilutions are included in the dataset because widely different masses of fuel were burned during each individual burn (see SI Table 2 for details) and some burns were diluted into the combustion room while others were diluted into the much more compact combustion stack. Despite this wide range of dilution conditions, the parameterization of optical properties with EC/(EC+OC) ratio appears robust.  It is important to note that this paper is not an attempt to say what the exact EC/OC ratio will be for a given fuel, as this may depend on dilution, but that if the EC/OC ratio is known at a given dilution then the optical properties can be predicted via the parameterizations presented.  Accordingly, the authors urge some caution in utilizing EC/OC emission factors from emissions that are not adequately diluted to predict regional optical properties (Akagi et al., 2011).

**Referee Comment:** A general comment on the paper is that a number of real-time properties of emissions were characterized at 1 Hz, but only 'burn-average' properties discussed. It would be interesting to see how these properties evolved for individual burns as it progressed, as in many cases there will be distinct phases with different properties, and the relative prevalence of these different properties may be very different for the same fuels in different conditions.

**Author Response:** We agree with the referee that optical properties are different at different phases of the burn, in fact they can be dramatically different during flaming vs. smoldering. During the measurements, we observed low SSA and AAE when the burn was dominated by flaming but high SSA and AAE when the burn was mostly smoldering.  Given that a burn might smolder for a very long time, but that the later-phases of this long smoldering may represent a small fraction of total emissions, it was decided that the most useful approach was to reported burn integrated values (summed absorption, summed extinction).  This allows us to compare room burns (which are naturally integrated) to stack measurements and gives the SSA of the sum of particles emitted during the burn. The focused of this paper is to parameterize SSA and AAE with MCE, EC/OC, and EC/(EC+OC). Since MCE and EC/OC were calculated as burn integrated, SSA and AAE were also reported in a similar manner.  While observing changing SSA with burn phase is indeed interesting, it is not directly relevant to this paper.

**Referee Comment:** It would be helpful if confidence intervals were provided on regression fits.

**Author Response:** We have added confidence intervals on regression fits

**Referee Comment:** One important detail left out of the final section, comparing results from the

(Yokelson et al. 2009) study, is that 'PM2.5' and 'BC' in this study were both determined optically (via nephelometer and PSAP, respectively). It is therefore a bit circular to use these to show that a parameterization based on chemical measurements can be used to represent optical properties. The 'calibration' of the PM/BC measurements in Yokelson et al does provide an indirect link to 'mass' measurements, but they are still optical measurements. This still may be a useful example of the applicability of your result, but needs to be used with proper caveats. To properly do uncertainty analysis on this, the uncertainty in the other assumptions (MAE, MSE) used to estimate BC and PM should also be included.

**Author Response:** We agree with the referee's comment. The purpose of this comparison was to test whether the parameterization can capture the effect of aging on SSA that was observed in different studies (Abel et al., 2003; Yokelson et al., 2009; Vakkari et al., 2014). Lack of availability of both EC/OC and SSA data of the aged biomass burning aerosol in these studies makes it difficult to check the performance of our parameterization during aging. This is why we used Yokelson et al. data even though PM/BC was determined optically. We have added following text in the document.

**Added Text Location:** section 3.5 after "instead of EC/(EC+OC) so several assumptions had to be made to implement our parameterization."

**Added Text:** One important note is that the $\Delta BC/\Delta PM_{2.5}$ reported in Yokelson et al. (2009) was derived from optical measurements.

**Referee Comment:** P3, L10-12 – This sentence is confusing. It makes it sound as if Indonesian Peat is the largest source of organic carbon on the ground (terrestrial). Also, combustion of peat is a varying source and I don't know if a statement so strong is justified.

**Author Response:** We modified the sentence to read, "Tropical peatlands are one of the largest reservoirs of terrestrial organic carbon".

**Referee Comment:** P4, L6 – Pretty sure you can't put Perma-pure in a canister this way? Did you use a Nafion dryer? Maybe thinking of some other compound?

**Author Response:** We change the sentence to read, "Dilution flow was generated from ambient air by passing it through an active-charcoal and permanganate (Purafil) scrubber to remove gas phase absorbers ($O_3$ and $NO_x$) followed by a HEPA filter to remove particulates".

**Referee Comment:** P4, L13-15 – This is a circular argument unless you have some a priori reason that the quantities you are comparing should have robust correlations.

**Author Response:** This entire paragraph has been modified to clarify statements about dilution.

**Referee Comment:** P5, L27 – clarify what is meant by excess

**Author Response:** We change the sentence to read, "background corrected" at P5, L27 and also

at P5, L30.

**Referee Comment:** P7, L4 – missing a word here, perhaps 'is'?

**Author Response:** We have modified the sentence, it now reads, "At high MCE, AAE is ~1 because BC dominates absorption"

**Referee Comment:** P8, L8-9 – It is not stated what the chosen functional form is, and why it was chosen.

**Author Response:** The power law function was chosen because it gave a good fit and was consistent with the power law function utilized to parameterize with MCE**.** We have modified to the text to read, "Figure 3 demonstrates that a power-law parameterization of SSA with EC/OC yields a function."

**Referee Comment:** P8, L20 – much of this strong correlation is driven by the fact that there are two clusters of data that are widely spread, through which a line can be drawn. An exponential-type curve could also be driven, and might asymptote at a more reasonable value as EC/TC goes to higher values.

**Author Response:** We appreciate the reviewer's comment, but we slightly disagree with the statement "much of this strong correlation is driven by the fact that there are two clusters of data that are widely spread, through which a line can be drawn". Since EC/TC and SSA can only vary from 0 to 1, EC/TC changes from 0.005 to 0.2 represent and SSA changes from 1 to 0.8 represent significant variations and the fit tracks the variations in SSA over this range of EC/TC well. Much of the atmospherically relevant biomass burning aerosol fall in this range of SSA. Additionally, if the X axis of the Figure 4 is plotted on a Log scale then the points spread out in a fashion very similar to Figure 3. We agree an exponential (or other functional forms) would also work but because the simple linear regression was equally accurate we did not proceed beyond this. Statistically, the simple model is best than any other complex form.

**Referee Comment:** P8, L24-26 – This sentence is awkward and difficult to understand.

**Author Response:** We have changed the sentence to read, "In fact, the robustness of the fits suggests that the EC/(EC+OC) ratio is able to predict the SSA and, to some extent, AAE even though information on particle size distribution, lensing, brown carbon and fuel types are not present, a rather surprising, but useful result."

**Referee Comment:** P 9, L15 – it would be helpful if some of this comparison were made graphically, either as a separate plot, or by adding some/all of these points on existing plots.

**Author Response:** We agree with the referee but the previous studies lack MCE and EC/OC data which prevents us from adding these points to existing plots.

**Referee Comment:** P9, L21-22 – Is this really all that can be said about this? If this is the case, it's really not clear whether it is worth including a table, especially in the main paper. If no systematic point can be made, put the table in the SI and just include a range of differences.

**Author Response:** We believe it is important to stress the differences in utilizing different parameterizations and believe it is important enough to leave the table intact. We have left the other wavelengths to the SI.

**Referee Comment:** P9, L23 – Extra word?

**Author Response:** We have removed the extra word.

**Referee Comment:** P9, L24-25 – Worth a discussion if you are saying your proposed parameterizations won't be applied in models. This is your original motivation – why would it only be used in this limited way if it is so much better than the alternatives?

**Author Response:** The statement will be changed. We will now say, "Because climate models need to mix different emission types, track SSA with extensive aging, and track particle losses, we anticipate that climate models will need parameterizations that include particle-size and refractive index and will not directly implement the parameterizations presented here. However, these parameterizations provide a critical tool to assess if a model implementation, based on assumptions about refractive index and coating thicknesses (Saleh et al., 2015), generates reasonable SSA estimates."

**Referee Comment:** P10, L12-13 – sentence fragment. This should be quantified: 'much worse'. EC-dominated combustion is more common in biofuel use, so may still be an issue.

**Author Response:** We change the sentence as "But the predicted values for "dark" plumes are consistently larger than measured values by about 35% on average."

**Referee Comment:** P10, L26 – Any suggestions as to why the AAEs determined for the same burns are so different?

**Author Response:** We suggest two possible reasons 1) We used different instrument to measure absorption coefficients which would potentially introduced different measurement uncertainties and 2) We estimated AAE from three wavelengths by liner fitting of log(absorption) vs log(wavelengths) while Liu et al calculated between 405/781. These two different method also introduced some variabilities in calculated AAE.

**Referee Comment:** P10, L27-28 – There should be a reference to a source with these data. Also, seems to be a missing word in this sentence.

**Author Response:** We change the sentence as "Other peats (North Carolina, Canadian) produced aerosol with similar optical properties to Indonesian peat (values can be found in Table have less impact on the global radiative budget".

**Referee Comment:** P11, L9 – Would be good to mention the physical significance of this intercept.

**Author Response:** We agree with the referee and added following text.
**Added Text:** "which signifies that in absence of EC, SSA due to OC is close to 1 for 532 and 660 nm while it is approximately 0.91 at 405 nm due to effect of brown carbon absorption at 405 nm."

**Referee Comment:** Table 5 –should be no more than 2 significant figures in % difference column

**Author Response:** We have modified the % difference column to 2 significant figure.

**References:**

Abel, S. J., Haywood, J. M., Highwood, E. J., Li, J., and Buseck, P. R.: Evolution of biomass burning aerosol properties from an agricultural fire in southern Africa, Geophys. Res. Lett., 30(15), 1783, doi:10.1029/2003GL017342, 2003.

Akagi, S. K., Yokelson, R. J., Wiedinmyer, C., Alvarado, M. J., Reid, J. S., Karl, T., Crounse, J. D. and Wennberg, P. O.: Emission factors for open and domestic biomass burning for use in atmospheric models, Atmos. Chem. Phys., 11, 4039–4072, doi:10.5194/acp-11-4039-2011, 2011.

Cheng, Y., He, K. B., Duan, F. K., Zheng, M., Ma, Y. L., and Tan, J. H.: Positive sampling artifact of carbonaceous aerosols and its influence on the thermal-optical split of OC/EC, Atmos. Chem. Phys., 9, 7243-7256, doi:10.5194/acp-9-7243-2009, 2009.

May, A. a., Levin, E. J. T., Hennigan, C. J., Riipinen, I., Lee, T., Collett, J. L., Jimenez, J. L., Kreidenweis, S. M. and Robinson, A. L.: Gas-particle partitioning of primary organic aerosol emissions: 3. Biomass burning, J. Geophys. Res. Atmos., 118, 11327–11338, doi:10.1002/jgrd.50828, 2013.

Vakkari, V., Kerminen, V.-M., Beukes, J. P., Titta, P., Zyl, P. G. van, Josipovic, M., Wnter, A. D., Jaars, K., Worsnop, D. R., Kulmala, M. and Laakso, L.: Rapid change in biomass burning aerosols by atmospheric oxidation, Geophys. Res. Lett., 2644–2651, doi:10.1002/2014GL059396, 2014.

Yokelson, R. J., Crounse, J. D., DeCarlo, P. F., Karl, T., Urbanski, S., Atlas, E., Campos, T., Shinozuka, Y., Kapustin, V., Clarke, A. D., Weinheimer, A., Knapp, D. J., Montzka, D. D., Holloway, J., Weibring, P., Flocke, F., Zheng, W., Toohey, D., Wennberg, P. O., Wiedinmyer,

C., Mauldin, L., Fried, A., Richter, D., Walega, J., Jimenez, J. L., Adachi, K., Buseck, P. R., Hall, S. R. and Shetter, R.: Emissions from biomass burning in the Yucatan, Atmos. Chem. Phys., 9, 5785–5812, doi:10.5194/acp-9-5785-2009, 2009.

---

## Author Comment (AC2) · 25 Jun 2016

We would like to thank the reviewer for their valuable suggestions and time. Our responses are given below. We believe the page and line numbers in the reviewer's comments were based on the manuscript that was initially submitted and was modified before publication in ACPD. This resulted in difference between line and page numbers in the comments and the current version of the manuscript in ACPD. Accordingly, we have removed the reviewer's line and page references and inserted the correct page and line numbers to avoid confusion.

**Anonymous Referee #1**

**Referee Comment:** Which parameter is more easily and accurately available for fires on regional and global scales? MCE or the amount of BC and BB OC (or OA)? Since OA/CO ratios for fires are quite variable and hard to predict in models, which SSA parameterization would then lead to the least uncertainty in the radiative effects of wildfire aerosols? See also my specific comment below.

**Author Response:** While published emission factors for biomass burning are relatively rare, both MCE and BC/OC (or OA) emissions factors are available (Akagi et al., 2011). The emission factors are vegetation (ecosystem) average values making the applicable to regional/global scales. The emission factors for BC/OA have been implemented in GEOS-CHEM (Saleh et al., 2015). While BC/OC (or OA) ratio is difficult to accurately predict in models, it is often easier to keep track of this ratio than MCE for aerosol because many models keep track of aerosol and gas-phase emissions separately. Neither BC/OC (or OA) or MCE can be accurately measured by satellites.

**Referee Comment:** MCE, EC/OC, and EC/(OC+EC) all have uncertainties/errors associated with them, so the fits are more appropriate if they're ODR with uncertainties as weights of the fit and not least-square linear regression lines as is done throughout the paper. (This is the main reason for my rating of the paper as 'major' revisions since all the figures, tables, and numbers in the text are to be updated accordingly).

**Author Response:** We agree with the referee that MCE, EC/OC, and EC/(EC+OC) all have uncertainties/error associated with them. Because we are comparing the predictive capabilities of a previously published MCE-based parameterization (that was done with least-squares regression) proposed by Liu et al. with our EC/OC based parameterization, we have decided to do most of the fitting with least-squares regression to compare apples to apples. Additionally, the fits for SSA vs. MCE and SSA vs. EC/OC are non-linear which makes applying least squares regression or ODR of fitting more complex. Given this, we have decided not to change the MCE or EC/OC fits to ODR, but we have made fits for SSA vs. EC/(EC+OC) with ODR and added the results to the SI. We also made a figure comparing the predicted SSA values based on SLR and ODR regression (as explained in section 3.5) and added this in the SI. The predicted SSA values based on SLR are slightly larger than ODR, but the difference is not statistically significant. A two-tailed p value of 0.748 when a two sample t test was performed with the results from both fits. We also add following text in section 3.2 and 3.5

**Added Text Location:** section 3.2 after the sentence "The y-intercepts of the fits….

**Added Text:** Fig. S1 shows that regression lines based on a simple liner regression (SLR) model and orthogonal distance regression (ODR) model are fairly similar at wavelengths of 660 and 532 nm for SSA vs. EC/(EC+OC). The regression lines for SSA vs. EC/(EC+OC) at 405 nm show significant deviations when fitted with the SLR vs. ODR methods, especially at higher EC/(EC+OC) ratio. This difference may be due to less data points in that region. Similarly, regression lines for AAE shows larger deviation between SLR and ODR methods at lower EC/(EC+OC) values, possibly due to less data points. ODR-based fits are provided for those who prefer this regression technique.

**Added Text Location:** section 3.5 after the sentence "This result suggests that the..

**Added Text:** Similarly, SSA values are predicted based on an ODR model and compared with the SSA predicted by an SLR model. Figure S2 shows the comparison of SSA predicted based on SLR vs. ODR models. The general trend shows that the predicted SSA based on SLR is higher than that based on ODR, but that difference is not statistically significant. We performed a two-tailed t test with the null hypothesis that predicted values are the same from both regression models and found a two-tailed p value of 0.748 at 532 nm.

**Referee Comment:** P4, L8: The authors discuss possible evaporation of semi-volatile components after sample dilution, but another factor is temperature differences in the relatively long sampling line. What was the temperature of the sampling lines kept at? If it wasn't controlled, how does this temperature difference impact redistribution of semi-volatile components of aerosols?

**Author Response:** The sampling line was at the room temperature of the lab and there was no temperature control. This is expected to have no impact on room-burn results because emissions were already cooled to room temperature. Stack emissions had also cooled to near-room temperature by the time they reached the top of the stack, though the exact temperature depends on the amount of biomass burned. The fact that fire integrated values of SSA and AAE from stack burns are nearly identical to the room burn values for similar burn conditions, strengthens the argument that neither dilution or the temperature of the sampling line significantly altered the optical properties.

**Referee Comment:** P4, L28: How is the response of activated carbon monolith to gaseous organic species? Could this sample treatment introduce negative artifacts in organic aerosols?

**Author Response:** It is possible that the activated carbon utilized to scrub $NO_x$ and ozone could introduce a small negative artifact by removing some volatile organics from the gas-phase and causing aerosol evaporation. However, this is expected to be a negligibly small effect given that the emissions were already diluted into the very large combustion room and had equilibrated to this highly-diluted environment. Again, the fact that nearly identical results were obtained for room-burn (highly diluted) and stack burns (less diluted) strongly suggests that the inlet is not introducing significant artifacts.

**Referee Comment:** P5, L27: What does 'excess' extinction and absorption mean? I'm not sure how SSA/AAE for stack burns in calculated- I thought it should be just based on average values of abs and ext, but not sure what summing up the 'excess' amounts mean.

**Author Response:** Excess means above background. We have changed the word "excess" to "background corrected" at P5, L27 and also at P5, L30. SSA/AAE were calculated based on the fire integrated absorption and extinction values during a stack burns. Fire-integrated means you add up all the absorption and extinction that occurred during the course of the burn. A straight time average is misleading because, in a stack burn, SSA and AAE vary by large amount from the flaming dominated part of burn to the smoldering dominated part. A fire may smolder for a long time, but a small fraction of mass emissions occurs during this time so an average value will be biased for those burns which have a shorter flaming period (few second) and a longer smoldering period (couple of minutes). This kind of measurement is common in instantaneous measurement during control burn in laboratory study (McMeeking et al., 2009; Liu et al., 2014; Stockwell et al., 2014).

**Referee Comment:** P6, Section 3.1: I agree that the data suggest EC/OC ratios are more variable for a given fuel than MCE and therefore relationships of optical properties with EC/OC are more robust. Can the authors elaborate on what specifically changes in the 'burn condition' that leads to such variability in aerosol characteristic? Is the different in the water content of the fuel or the starting temperature of the fire, etc. etc?

Author Response: There are a large number of parameters that determine the EC/OC ratio for a given burn. Some parameters are the surface area to mass of the fuel (stick vs. log vs. pine needle), the way the fuel is stacked or layered, the moisture content of the fuel, and the nature of the fuel (grass vs. wood). Our goal in this paper is not to describe what caused a given EC/OC ratio, for that type of information one must reference an emission factor paper such as Akagi et al., 2011.

**Referee Comment:** P9, Section 3.4: Can you perform a simple calculation to estimate instantaneous TOA forcing difference of BB aerosol depending on the choice of SSA (new vs. old parameterization) so readers get an idea about the magnitude of the change in forcing? This calculation should be done considering the uncertainties in MCE, EC/OC (or EC/(OA+EC)) and SSA.

**Author Response:** Unfortunately, there is no easy way to do a simple calculation of this type. Given that different fuels have different EC/OC emissions and that the EC/OC emission depends on moisture etc., one would need an inventory of global fuels, to couple this inventory to the amount of burning that occurs in different regions, then inject the particles in the atmosphere at the appropriate heights and remove them at appropriate rates. This is a complex problem that requires a global model of some type. In general, if SSA drop from 1 to 0.9, TOA forcing can drop by 50 to 100% depending upon the surface albedo (Russell et al., 2002). Saleh et al. (2015) have recently calculated that different parameterizations can significantly change the TOA forcing from biomass burning utilizing GEOS-Chem.

**Referee Comment:** P9, L24: I'm confused by the statement "While climate models may not directly parameterize optical properties based on EC/OC, the parameterization provides a good sanity check of model schemes to predict optical properties." If models have EC/OC data to do this sanity check, do the authors not recommend the modelers to use this parameterization instead of other estimates? If yes, I think the sentence needs to be rephrased. If not, elaborate why this shouldn't be the recommended approach.

**Author Response:** The statement will be changed. We will now say, "Because climate models need to mix different emission types, track SSA with extensive aging, and track particle losses, we anticipate that climate models will need parameterizations that include particle-size and refractive index and will not directly implement the parameterizations presented here. However, these parameterizations provide a critical tool to assess if a model implementation, based on assumptions about refractive index and coating thicknesses (Saleh et al., 2015), generates reasonable SSA estimates."

**Referee Comment:** P10, L21-24: I do understand that the average SSA of peat burning aerosol is lower, but given the uncertainty for the SSA values, the difference at 532 nm vs. 405 nm is not really significant. The relatively high AAE is more convincing for the presence of BrC in peat burning. What is the uncertainty in AAE for this sample? Add that value as well.

**Author Response:** The uncertainty for the AAE has been added. In terms of uncertainty in SSA, we have modified the error stated to be the error in the mean rather than the error of an individual measurement, which was originally quoted. This makes the difference between 532 and 405 nm more significant.

**Referee Comment:** P10, Line4: I question the assumption of PM2.5 in a fire being composed of only BC and OM. In most fires, there could also be aerosol nitrate and chloride. How does the ratio of EC/(EC+OC), and therefore, estimate of SSA change if say 5-15% of PM2.5 is assumed to be inorganics? Also, looking at Table 3 in Yokelson et al., ACP 2009, there were direct PM1-OM measurements. Why not use that measure of OM when calculating OC?

**Author Response:** We agree with the referee that $PM_{2.5}$ in a fire is not composed of only BC and OM. We have modified our calculation to assume 39 % of $PM_{2.5}$ is OC as estimated by Yokelson et al. (2009) during the study. We have corrected the text and updated Fig. 5 with this assumption. The parameterization performs well with this new assumption and no further adjustments to the text were needed.

Added text location: Section 3.5

**Added Text:** $\Delta BC/\Delta PM_{2.5}$ was converted in to EC/(EC+OC) by setting the OC mass fraction to 39 ±9 % of the $PM_{2.5}$ as stated by Yokelson et al. (2009). This calculation assumes BC and EC mass are identical.

**Referee Comment:** Additionally, BC and EC are not necessarily presenting the same type of species. Can you reference papers that perform both measurements on a series of burned fuels and comment on the 'goodness' of this assumption and how it will impact the predicted SSA?

**Author Response:** We agree BC and EC are not necessarily presenting the same type of species. But EC is often used as surrogate of BC. We have cited Salako et al. (2012) in our manuscript which state that, for a burn with 82 % biomass burning emissions, 17% diesel emissions, and 1% other, BC = 1.06*EC with $R^2 = 0.91$. Based on this, our EC/(EC+OC) parameterization and the predicted SSA and AAE will be not significantly different if BC is utilized instead of EC.

**Minor Comments:**

**Referee Comment:** P2, L7 primary OA=POA

**Author Response:** We have changed primary OA to POA

**Referee Comment:** P3,L16: 'effectiveness of our . . .''

**Author Response:** We have changed the sentence to, "We also show that predicted SSA based on the EC/(EC+OC) parameterization is similar to measured SSA during the first few hours of aging from the Yucatan peninsula in Mexico (Yokelson et al., 2009)."

**Referee Comment:** P7, L4-5 "At high MCE, AAE is ~1 because BC absorption proportional to frequency" incomplete sentence. Also, by frequency, do you mean 'wavelength'?

**Author Response:** We have modified the sentence, it now reads, "At high MCE, AAE is ~1 because BC dominates absorption"

**Referee Comment:** P7, L5-6: should clarify that low MCE burns give high SSA at long wavelengths, since in the following sentences it's mentioned that the OA in low MCE burns is highly absorbing as BrC.

**Author Response:** We have changed the sentence to, "In contrast, fuels that burn with low MCE are dominated by OC emissions, which predominantly scatter light at long wavelengths resulting in SSA values nearing unity at 532 and 660 nm and larger values of AAE".

**Referee Comment:** P9, L18: consider ". . ..how significant of an impact. . ."
**Author Response:** We have modified the sentence as suggested by referee.

**Referee Comment:** P9, L23: during . . .? Incomplete sentence

**Author Response:** We have deleted the extra word "during".

**Referee Comment:** P10: Indonesian Peat section should be 3.6

**Author Response:** Indonesian Peat section in now 3.6

**Referee Comment:** P10, L12: What's the explanation for the parameterization not capturing the measured SSA in plumes with lower SSA? Also, start the sentence with "However" instead of "But", or combine the two sentences

**Author Response:** This could be a limitation of our parameterization to parameterize burns with extremely high EC (EC/OC much greater than unity). Vakkari et al note that these "dark" plumes are rarely observed in the atmosphere. We change the sentence as "But the predicted values for "dark" plumes are consistently larger than measured values by about 35% on average.

**Referee Comment:** P10, L27-28: Rephrase the sentence, sounds like summary bullets and not a complete sentence.

**Author Response:** We change the sentence as "Other peats (North Carolina, Canadian) produced aerosol with similar optical properties to Indonesian peat (values can be found in Table) have less impact on the global radiative budget.

**Referee Comment:** P11, L2: When giving the range of SSA, separate it for 405 nm vs. 660 nm.

**Author Response:** We now report SSA ranges for both 405 nm and 660 nm.

**Referee Comment:** P11, L11-12: Indicate again the errors bars for SSA of peat aerosol at 405 nm and longer wavelengths as well as the error bar for AAE.

**Author Response:** Errors bars are included for peat aerosol on P11, line 11-12.

**References**:

Akagi, S. K., Yokelson, R. J., Wiedinmyer, C., Alvarado, M. J., Reid, J. S., Karl, T., Crounse, J. D. and Wennberg, P. O.: Emission factors for open and domestic biomass burning for use in atmospheric models, Atmos. Chem. Phys., 11, 4039–4072, doi:10.5194/acp-11-4039-2011, 2011.

Liu, S., Aiken, A. C., Arata, C., Dubey, M. K., Stockwell, C. E., Yokelson, R. J., Stone, E. a, Jayarathne, T., Robinson, A. L., Demott, P. J. and Kreidenweis, S. M.: Aerosol single scattering albedo dependence on biomass combustion efficiency: Laboratory and field studies, Geophys. Res. Lett., 41, 742–748, doi:10.1002/2013GL058392, 2014.

McMeeking, G. R., Kreidenweis, S. M., Baker, S., Carrico, C. M., Chow, J. C., Collett, J. L., Hao, W. M., Holden, A. S., Kirchstetter, T. W., Malm, W. C., Moosmüller, H., Sullivan, A. P. and Cyle E., W.: Emissions of trace gases and aerosols during the open combustion of biomass in the laboratory, J. Geophys. Res., 114, D19210, doi:10.1029/2009JD011836, 2009.

Russell, P. B., Redemann, J., Schmid, B., Bergstrom, R. W., Livingston, J. M., McIntosh, D. M., Ramirez, S. a., Hartley, S., Hobbs, P. V., Quinn, P. K., Carrico, C. M., Rood, M. J., Öström, E., Noone, K. J., von Hoyningen-Huene, W. and Remer, L.: Comparison of Aerosol Single Scattering Albedos Derived by Diverse Techniques in Two North Atlantic Experiments, J. Atmos. Sci., 59(3), 609–619, 2002.

Salako, G. O., Hopke, P. K., Cohen, D. D., Begum, B. A., Biswas, S. K., Pandit, G. G., Chung, Y. S., Rahman, S. A., Hamzah, M. S., Davy, P., Markwitz, A., Shagjjamba, D., Lodoysamba, S., Wimolwattanapun, W. and Bunprapob, S.: Exploring the variation between EC and BC in a variety of locations, Aerosol Air Qual. Res., 12, 1–7, doi:10.4209/aaqr.2011.09.0150, 2012.

Saleh, R., Marks, M., Heo, J., Adams, P. J., Donahue, N. M. and Robinson, A. L.: Contribution of brown carbon and lensing to the direct radiative effect of carbonaceous aerosols from biomass and biofuel burning emissions, J. Geophys. Res. Atmos., 120, doi:10.1002/2015JD023697-T, 2015.

Stockwell, C. E., Yokelson, R. J., Kreidenweis, S. M., Robinson, A. L., DeMott, P. J., Sullivan, R. C., Reardon, J., Ryan, K. C., Griffith, D. W. T. and Stevens, L.: Trace gas emissions from combustion of peat, crop residue, biofuels, grasses, and other fuels: configuration and FTIR component of the fourth Fire Lab at Missoula Experiment (FLAME-4), Atmos. Chem. Phys., 14, 9727–9754, doi:10.5194/acp-14-9227-2014, 2014.

Yokelson, R. J., Crounse, J. D., DeCarlo, P. F., Karl, T., Urbanski, S., Atlas, E., Campos, T., Shinozuka, Y., Kapustin, V., Clarke, A. D., Weinheimer, A., Knapp, D. J., Montzka, D. D., Holloway, J., Weibring, P., Flocke, F., Zheng, W., Toohey, D., Wennberg, P. O., Wiedinmyer, C., Mauldin, L., Fried, A., Richter, D., Walega, J., Jimenez, J. L., Adachi, K., Buseck, P. R., Hall, S. R. and Shetter, R.: Emissions from biomass burning in the Yucatan, Atmos. Chem. Phys., 9, 5785–5812, doi:10.5194/acp-9-5785-2009, 2009.

---

## Author Comment (AC3) · 25 Jun 2016

**We would like to thank the reviewer for their valuable suggestions and time. Our responses are given below.**

Anonymous Referee #4

**Referee Comment:** Pg 3 Ln 25-26. The authors explain about observation of fire transitions from flaming to smoldering. It would be good to add information or clarify how was this done. Also how fractions of flaming to smoldering (Pg 5 Ln 28-30) were maintained during different experiments.
**Author Response:** We have not attempted to control transitions from flaming to smoldering. Once a fire was ignited, it was left to burn. Fire integrated absorption and extinction were utilized to avoid over-emphasizing properties of the smoldering or flaming portion of the burn.

**Referee Comment:** Pg 7 Ln 29. This statement about EC/OC depends more on burn conditions than fuel type can mislead. As it is shown in this study (Fig. 2), for certain fuels, such as peat, the different combustion types do not change EC/OC significantly. Therefore, fuel types do influence the EC/OC ratio.

**Author Response:** We have altered the sentence to read, "Furthermore, this data demonstrates that EC/OC depends significantly on burn conditions in addition to fuel type."

**Referee Comment:** Pg 8 Ln 24-26. The authors suggest that EC/(EC+OC) ratio is able to predict AAE. I think the r of the least square fit is not that strong for this case (r=-0.79) different from fitting for SSA. The statement may need to be revised.

**Author Response:** We agree with the referee that r value for AAE fit is not that strong. We have modified to the sentence to read, "In fact, the robustness of the fits suggests that the EC/(EC+OC) ratio is able to predict the SSA and, to some extent, AAE even though information on particle size distribution, lensing, brown carbon, and fuel types are not present, a rather surprising, but useful result."

**Referee Comment:** Pg 8 Ln 29-30. I think the RMSE values from fit of this study and fit by Liu et al. (Table 3) are not similar. They are about 20-30% different. The statement may need to be revised.
**Author Response:** The sentence has been modified to read, "For the MCE approach, the RMSE is similar whether coefficients from a least squares fit to our data are used or whether the coefficients proposed by Liu et al. (2014) are used, though the error is slightly lower when the coefficients from the fit to our data are used."

Technical comments
**Referee Comment:** Pg 4 Ln 6. What does it mean "a canister filled with Perma-Pure"? Perma-Pure is a manufacturer name.

**Author Response:** We change the sentence to read, "Dilution flow was generated from ambient air by passing it through an active-charcoal and permanganate (Purafil) scrubber to remove gas

phase absorbers ($O_3$ and $NO_x$) followed by a HEPA filter to remove particulates".

**Referee Comment:** Pg 6 Ln 14. What is temperature of filter storage?

**Author Response:** We have rewritten the sentence to include the temperature of filter storage. It now read, "Filters were stored in clean aluminum foil-lined petri dishes sealed with Teflon tape, and stored frozen ($-20^0$ C) before and after the analysis".

**Referee Comment:** Pg 7 Ln 18-19. I am confused with this statement. Liu et al. provided parameterization only for 405 and 532 nm, and on Fig. 1, there is no black fitted line for panel C (660 nm). So what does 660 nm refer to?

**Author Response:** We have deleted the sentence "At 532 and 660 nm, there are also notable errors at low MCE."

**Referee Comment:** Pg 9 Ln 22-23. This sentence is not finished?

**Author Response:** We have removed the extra word.

**Referee Comment:** Pg 10 Ln 10. Provide correlation plot and value of SSA at 660 nm and at 637 nm in SI.
**Author Response:** We were not able to obtain exact values of SSA at 637 even though we contacted Vakkari et al. The analysis is based on close inspection of their published figures with a program that converts figures to numerical values. We would not feel comfortable publishing exact values without the approval of Vakkari et al.

**Referee Comment:** Pg 10 Ln 22-25. In which table we can find the SSA and AAE for Indonesian peat and the other fuels?

**Author Response:** SSA and AAE values for all burns are available in SI (Table S2).

**References:**

Liu, S., Aiken, A. C., Arata, C., Dubey, M. K., Stockwell, C. E., Yokelson, R. J., Stone, E. a, Jayarathne, T., Robinson, A. L., Demott, P. J. and Kreidenweis, S. M.: Aerosol single scattering albedo dependence on biomass combustion efficiency: Laboratory and field studies, Geophys. Res. Lett., 41, 742–748, doi:10.1002/2013GL058392, 2014.

Vakkari, V., Kerminen, V.-M., Beukes, J. P., Titta, P., Zyl, P. G. van, Josipovic, M., Wnter, A. D., Jaars, K., Worsnop, D. R., Kulmala, M. and Laakso, L.: Rapid change in biomass burning aerosols by atmospheric oxidation, Geophys. Res. Lett., 2644–2651, doi:10.1002/2014GL059396, 2014.

---

## Author Comment (AC4) · 25 Jun 2016

We would like to thank the reviewer for their valuable suggestions and time. Our responses are given below. We believe the page and line numbers in the reviewer's comments were based on the manuscript that was initially submitted and was modified before publication in ACPD. This resulted in difference between line and page numbers in the comments and the current version of the manuscript in ACPD. Accordingly, we have removed the reviewer's line and page references and inserted the correct page and line numbers to avoid confusion.

**Anonymous Referee #2**

**Referee Comment:** My main question at this point relates to their choice of r as the figure of merit in their (non-linear) fitting.

**Author Response:** We agree that r is not a good parameter to assess a non-linear fit. This is why we have given RMSE in Table 3, which we believe is a better parameter for the non-linear fits. We have given r for the non-linear fits because it has been stated in previous publications with which we are comparing. We do believe r is relevant for the EC/(EC+OC) parameterization, which is linear.

**Referee Comment:** P1, L17: Suggest replacing "significant" with "substantial" or "important" so as not to imply statistical significance.
**Author Response:** We have changed the word "significant" to "substantial".

**Referee Comment:** P1, L18: Suggest replacing "inferred" with "predicted".
**Author Response:** We have changed the word "inferred" to "predicted".

**Referee Comment:** P1, L20: I find ": : :emission factors for the MCE: : :" to be unclear. EFs of what?
**Author Response:** We have changed the sentence to read, "It has been suggested that SSA can be effectively parameterized via the modified combustion efficiency (MCE) of a biomass-burning event and that this would be useful because emission factors for CO and $CO_2$ from which MCE can be calculated are available for a large number of fuels".

**Referee Comment:** P1, L27 and General Question: Pearson's r is a parameter that describes the linear correlation between two variables. Here, it seems to be applied to one data set that is linearly related (SSA vs. EC/(EC+OC)) and two that are not (SSA vs. MCE and SSA vs. EC/OC). Thus, are the r values really comparable? What does an r value mean for a non-linear relationship? Might a different statistical test be applied? Perhaps Spearman's or Kendall's rank correlation coefficients or a Pearson's Chi Square test?

**Author Response:** We agree with the referee that Pearson's r gives the measure of linear dependency between two variables and is not very meaningful for nonlinear relationship. However, a key focus of this paper is to compare the predictive capabilities of MCE and EC/OC. A MCE parameterization for SSA (nonlinear relation with SSA) was already published (Liu et al., 2014) and concluded that based on $R^2$ value that MCE can explain 60% in variability in SSA. To

show EC/OC is better than MCE we also made r value comparison because the article that we are comparing uses r values. Because we recognize the fault in utilizing r or $R^2$, we also compare the root mean square error (RMSE) of different approaches which is much more useful in comparing the model predictive capabilities. Even though, r value for nonlinear relationship is a rather poor assessment of fit, there are numerous publications that give this result in recent years too (Liu et al., 2014; Lu et al., 2015; Cui et al., 2016).

**Referee Comment:** P2, L9: The authors cite Stier (2007) as evidence that "most climate models treat organic carbon as purely scattering." However, it is clear from Fig. 1 in Stier (2007) that the OC is somewhat absorbing throughout the visible. In fact, most models treat OC as slightly absorbing (see e.g. the OPAC database).

**Author Response:** We appreciate the referee's effort in pointing out this improperly cited source and overstatement. We changed the sentence as "Although some climate model treat organic carbon (OC) as purely scattering (Myhre G et al., 2007), OC….".

**Referee Comment:** P2, L13: The inclusion of the reference to Washenfelder here seems quite selective, as there are many regions where biomass burning has been implicated as a source of BrC. Not that it is not a nice study, but is there a reason why this study is being highlighted?

**Author Response:** We agree with the referee that biomass burning has been implicated as a source of BrC. But there is significant uncertainty regarding the relative contribution of BrC by different sources (biomass burning, SOA, fossil fuels) in many regions. Washenfelder et al. found that biomass burning is the dominant source of BrC in southeastern US this is the reason why this study is highlighted.

**Referee Comment:** P2, L17: I suggest that the Saleh reference is removed and only the Feng (global model result) reference is retained.
**Author Response:** Reference is removed.

**Referee Comment:** P2, L23: What is meant by SSA and AAE are "commonly implemented in models"? Models don't specify SSA. Similarly, what is meant by "SSA and AAE are also critical for satellite retrievals"? Critical for or are important retrieved information from?

**Author Response:** The sentence has been modified to read, "Single scattering albedo (SSA) and absorption angstrom exponent (AAE) are commonly used parameters that contain the necessary information on aerosol absorption and scattering to calculate radiative effects".

**Referee Comment:** P3, L9: No reference to Salako et al. is provided. Also, I would contend that this really remains to be demonstrated as "charring" is known to be a particularly important for

biomass burning. Also, the authors might compare their longest wavelength denuded-particle absorption measurements to the EC measurements to argue that there is a reasonable relationship between BC and EC for this data set.

**Author Response:** We have inserted the reference to Salako et al. We agree there is some difficulty in equating EC to BC and this is something that needs to be further examined. Because converting our 660 nm absorption to BC would require assuming a MAC, we do not pursue this.

**Referee Comment:** P7, L4: The sentence starting "At high MCE" is a fragment.

**Author Response:** We have modified the sentence, it now reads, "At high MCE, AAE is ~1 because BC dominates absorption

**Referee Comment:** Fig. 1: It seems odd that the least squares fit (red line) doesn't match the data at smaller MCE values at 660 nm (most notably). Is there a reason for this? The functional form used (which should be given in the main text as well) should allow for better agreement at these low MCE values. Also, it is unclear if the fits were performed with/without accounting for the uncertainties in the individual points.

**Author Response:** A discussion of the reason for the mismatch between fit and data at smaller MCE has been added on Page 7 line 19.

**Referee Comment:** Fig. 3: The fits the authors retrieve allow for unphysical SSA values > 1. I suggest that they redo their fits, constraining the maximum retrievable SSA to be <= 1. This amounts to constraining the k0 in their fit equation to be <= 1. This links to P8,L14, where the authors note that this fitting does lead to SSA values > 1. But this is a solvable problem. Physical realism can be imposed on the fits.

**Author Response:** This issue has been corrected by constraining the fits to have a maximum SSA of 1. Text has been added on Page 8 line 14.

**Referee Comment:** P8, L5: the authors might indicate what they consider the EC/OC value at which composition is "dominated" by EC.

**Author Response:** The sentence has been modified to read, "When the composition has equal or more EC than OC, there is less dependence of SSA on wavelength and AAE values are less than 2".

**Referee Comment:** P8, L9: horizontal should be vertical.

**Author Response:** Horizontal has been changed to vertical.

**Referee Comment:** P8, L17: Are the data truly more "bunched" or is the difference that Fig. 4 uses a linear scale and Fig. 3 a log scale for the x-axis? I think the latter.

**Author Response:** We agree with the referee that the effect is due to the use of linear vs log scale. We removed the sentence starting "The simple nature of the linear………"

**Referee Comment:** P8, General: The authors discuss the robustness of their fits and the ability of EC/(EC+OC) to be used as a predictor. Although I generally agree, a few thoughts: (i) I think that the authors are overstating the case for AAE, as the correlation coefficient is only 0.79. (ii) Regarding the 405 nm measurements, yes, the fit gets a <1 SSA value when EC/(EC+OC) = 0. But it is also clear that the zero intercept here differs substantially from the data points. In other words, the fit is certainly "good" but the model fit and observed SSA values differ by 0.03 or more, which is small yet non negligible. (iii) Can the authors include confidence bands?

**Author Response:** (i) We agree with the referee that fitting with AAE is not very good. All we mean to say is that we can predict SSA and AAE with the value of EC and OC without the need to reference other properties like size distribution, lensing effect, presence of BrC. We believe including the r for the AAE fit is adequate to show that the fit is significantly weaker than that for SSA. (ii) we absolutely agree with the referee that there is uncertainty here. (iii) a 95% CI for fitting is included for the parameters of the fit. We feel adding the 95% confidence intervals to the plot would make the plot too busy.

**Referee Comment:** Table 5 and discussion: Do the MCE and EC/OC from the literature for biomass burning emissions generally agree with the observations here in terms of functional form?
**Author Response:** The values of MCE and BC/OC values from the literature are in the same range that we observed in this study. Also, the nature of BC/OC vs MCE from the literature follow a similar pattern to that shown in Fig. 2 of the main text).

**Referee Comment:** P9, L24: I find the meaning of the following sentence to be unclear: "While climate models may not directly parameterize optical properties based on EC/OC, the parameterization provides a good sanity check of model schemes to predict optical properties." Can the authors clarify how this table and discussion provides a "sanity check"?
**Author Response:** The statement will be changed. We will now say, "Because climate models need to mix different emission types, track SSA with extensive aging, and track particle losses, we anticipate that climate models will need parameterizations that include particle-size and refractive index and will not directly implement the parameterizations presented here. However, these parameterizations provide a critical tool to assess if a model implementation, based on assumptions about refractive index and coating thicknesses (Saleh et al., 2015), generates reasonable SSA estimates."

**Referee Comment:** P10, L10: What is meant by "reasonably good?" As good as the case that is shown? Can this just be shown?

**Author Response:** We state "the predictions are within roughly ± 5%..." which is what we describe to be reasonably good. We were not able to obtain exact values of SSA at 637 even though we contacted Vakkari et al. The analysis is based on close inspection of their published figures with a program that converts figures to numerical values. We would not feel comfortable publishing exact values without the approval of Vakkari et al.

**Referee Comment:** P10, L18: If the peat burning was unintentional and a result of e.g. drought, I suggest the authors say "through unintentional peat burning."

**Author Response:** Indonesian peat burning are mostly anthropogenic (Bompard et al., 1999) but the article that we cited was based on 1997 El Nino event.

**Referee Comment:** P7, L6: To set things up for later in the paper, the authors might report the mean value for peat here in addition to the maximum. Some discussion of the variability would also be helpful (later in section 3.4).

**Author Response:** We believe it is most efficient to state this once in Section 3.4 that discusses Indonesian Peat.

**Referee Comment:** P10, L27: this is a sentence fragment.

**Author Response:** We change the sentence as "Other peats (North Carolina, Canadian) produced aerosol with similar optical properties to Indonesian peat (values can be found in Table have less impact on the global radiative budget".

**Referee Comment:** General: I suggest that the authors adopt the terminology "aerosol particles" through-out much of the particle, to indicate that they are looking at the particulate matter and not the associated gaseous material.

**Author Response:** The title of the article clearly states that this article is about aerosol emissions. We have also modified the first sentence of the results and discussion to read, "Single scattering albedo (SSA) and absorption angstrom exponent (AAE) of aerosol emissions were measured during 41 individual burns of twelve different fuels during FLAME-4

**References:**

Bompard, J. M. & Guizol, P.: Land Management in South Sumatra Province, Indonesia. Fanning the Flames: The Institutional Cause of Vegetation Fires (European Union Forest Fire Prevention and Control Project and IndonesianMinistry of Forestry and Estate Crops, Jakarta, 1999).

Cui, X., Wang, X., Yang, L., Chen, B., Chen, J., Andersson, A. and Gustafsson, Ö.: Radiative absorption enhancement from coatings on black carbon aerosols, Sci. Total Environ., 551-552, 51–56, doi: 10.1016/j.scitotenv.2016.02.026, 2016.

Liu, S., Aiken, A. C., Arata, C., Dubey, M. K., Stockwell, C. E., Yokelson, R. J., Stone, E. a, Jayarathne, T., Robinson, A. L., Demott, P. J. and Kreidenweis, S. M.: Aerosol single scattering albedo dependence on biomass combustion efficiency: Laboratory and field studies, Geophys. Res. Lett., 41, 742–748, doi:10.1002/2013GL058392, 2014.

Lu, Z., Streets, D. G., Winijkul, E., Yan, F., Chen, Y., Bond, T. C., Feng, Y., Dubey, M. K., Liu, S., Pinto, J. P. and Carmichael, G. R.: Light Absorption Properties and Radiative Effects of Primary Organic Aerosol Emissions, Environ. Sci. Technol., 49, 4868–4877, doi: 10.1021/acs.est.5b00211, 2015.

Saleh, R., Marks, M., Heo, J., Adams, P. J., Donahue, N. M. and Robinson, A. L.: Contribution of brown carbon and lensing to the direct radiative effect of carbonaceous aerosols from biomass and biofuel burning emissions, J. Geophys. Res.  Atmos., 120, doi:10.1002/2015JD023697-T, 2015.

Vakkari, V., Kerminen, V.-M., Beukes, J. P., Titta, P., Zyl, P. G. van, Josipovic, M., Wnter, A. D., Jaars, K., Worsnop, D. R., Kulmala, M. and Laakso, L.: Rapid change in biomass burning aerosols by atmospheric oxidation, Geophys. Res. Lett., 2644–2651, doi:10.1002/2014GL059396, 2014.

---

## Author Response (AR2)

**We would like to thank Co-editor for his valuable suggestions and time. We have accepted all the minor edits.**

Comment: Abstract, Line 21: Remove "are available" at the end of this sentence.

Author Response: "are available" at the end of the sentence is removed.

Comment: Section 2.1, Page 4, Line 10: Change "weather" to "whether".

Author Response: We have changed "weather" to "whether".

Comment: Section 3.6, Page 11, Line 18: Add table number after word "Table".

Author Response: We have added the table number.

Comment: Section 3.6, Page 11, Line 18: After "Table)" add ", but"

Author Response: We have added ", but" after "Table)".

Comment: Conclusions, Page 11, Line 31: Change "Which" to "This"

Author Response: "Which" is changed to "This"